# Towards Effective and Interpretable Human-AI Collaboration in MOBA Games

## Abstract

MOBA games, e.g., *Dota2* and *Honor of Kings*, have been actively used as the testbed for the recent AI research on games, and various AI systems have been developed at the human level so far. However, these AI systems merely focus on how to compete with humans, less exploring how to collaborate with humans. To this end, this paper makes the first attempt to investigate human-AI collaboration in MOBA games. In this paper, we propose to enable humans and agents to collaborate through explicit communications by designing an efficient and interpretable **M**eta-**C**ommand **C**ommunication-based framework, dubbed MCC, for accomplishing effective human-AI collaboration in MOBA games. The MCC framework consists of two pivotal modules: 1) an interpretable communication protocol, i.e., the Meta-Command, to bridge the communication gap between humans and agents; 2) a meta-command value estimation model, i.e., the Meta-Command Selector, to select a valuable meta-command for each agent to achieve effective human-AI collaboration. Experimental results in *Honor of Kings* demonstrate that MCC agents can collaborate reasonably well with human teammates and even generalize to collaborate with different levels and numbers of human teammates. Videos are available at https://sites.google.com/view/mcc-demo.

## 1 Introduction

Games, as the microcosm of real-world problems, have been widely used as testbeds to evaluate the performance of Artificial Intelligence (AI) techniques for decades. Recently, many researchers focus on developing various human-level AI systems for complex games, such as board games like *Go* [27, 28], First-Person Shooting (FPS) games like *ViZDoom* [14], Real-Time Strategy (RTS) games like *StarCraft 2* [34], and Multi-player Online Battle Arena (MOBA) games like *Dota 2* [22]. However, these AI systems focus merely on how to compete instead of collaborating with humans, leaving Human-AI Collaboration (HAC) in complex environments still to be investigated.

In this paper, we study the HAC problem in complex MOBA games, which is characterized by multi-agent cooperation and competition mechanisms, long time horizons, enormous state-action spaces ($10^{20000}$), and imperfect information [22, 26, 38]. HAC requires the agent to collaborate reasonably with various human teammates. One straightforward approach is to improve the generalization of agents, that is, to collaborate with an enough diverse population of teammates during training. There are some Population-Based Training (PBT) based algorithms and learning systems [1, 2, 10, 11, 31, 41] proposed to improve the generalization of agents in video games by constructing a diverse population of agents in different ways. However, this approach requires a vast amount of diverse data and massive computing resources, posing a big computational obstacle for complex MOBA games.

Human team success in MOBA games requires not only subtle individual micro-operations but also excellent communications and collaborations among teammates on macro-strategies, i.e., long-term intentions [8, 37]. Consequently, we focus on enabling humans and agents to collaborate through

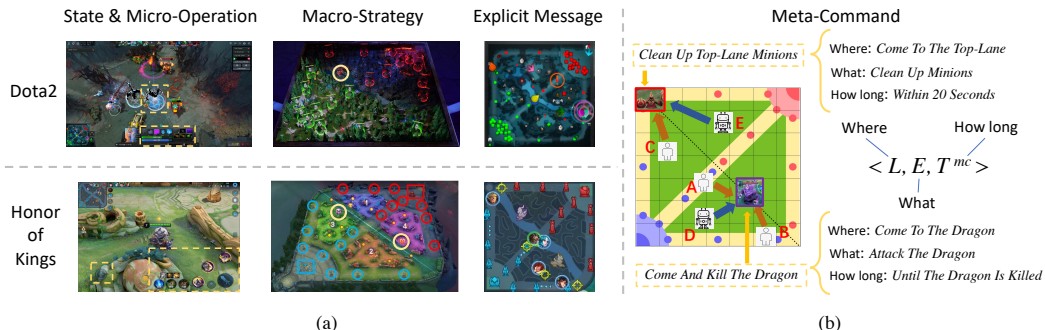

Figure 1: **MOBA game-related introduction.** (a) Key elements of MOBA games such as *Dota 2*, *Honor of Kings*, etc. Players observe from the *state* of the environment, make *micro-operations* and *macro-strategies* decisions, and collaborate through *explicit messages* (e.g.,text and signals). (b) Example of collaboration via meta-commands. The *Come And Kill The Dragon* is more valuable for humans A and B and agent D to collaborate, while the *Clean Up Top-Lane Minions* is more valuable for human C and agent E to collaborate.

explicit communications and propose an efficient and interpretable Meta-Command Communication-based human-AI collaboration framework, dubbed MCC, to solve the HAC problem in MOBA games. First, we design an interpretable communication protocol, i.e., the Meta-Command, as a general representation of macro-strategies to bridge the communication gap between agents and humans. Both macro-strategies sent by humans and messages outputted by agents can be converted into unified meta-commands (see Figure 1). Second, following Gao *et al.* [8], we construct a hierarchical model that includes the command encoding network (macro-strategy layer) and the meta-command conditioned action network (micro-action layer), used for agents to generate and execute meta-commands, respectively. Third, we propose a meta-command value estimation model, i.e., the Meta-Command Selector, to select the optimal meta-command for each agent to execute. The training process of the MCC framework consists of three phases. We first train the command encoding network to learn the distribution of meta-commands sent from humans. Afterward, we train the meta-command conditioned action network to ensure that the agent has the near-human completion rate for meta-commands. Finally, we train the meta-command selector to ensure that the agent can select a valuable meta-command to achieve effective collaboration. We train and evaluate the agent in *Honor of Kings* 5v5 mode with a full hero pool (over 100 heroes). Experimental results demonstrate the effectiveness of the MCC framework. In general, our contributions are as follows:

- To the best of our knowledge, we are the first to investigate the HAC problem in MOBA games. We propose an efficient and interpretable Meta-Command Communication-based framework dubbed MCC to achieve effective human-AI collaboration in MOBA games.

- We design an interpretable communication protocol to bridge the communication gap between humans and agents. In addition, we propose a meta-command value estimation model to select a valuable meta-command for each agent to achieve effective human-AI collaboration.

- We introduce the training process of the MCC framework in a typical MOBA game *Honor of Kings* and evaluate it in practical human-AI game tests. Experimental results show that MCC agents can reasonably collaborate with different levels and numbers of human teammates.

## 2 Related Work

### 2.1 MOBA Games AI Research

MOBA games, such as *Dota 2* and *Honor of Kings*, have attracted much attention from AI researchers due to their multi-agent cooperative and competitive mechanics, long time horizons, partial observation, and enormous state-action spaces [22, 38]. Recently, OpenAI *et al.* [22] introduced an AI system named OpenAI-Five that defeated professional players in *Dota 2* 5v5 mode under the condition of limited heroes. Ye *et al.* [38, 39, 40] proposed another learning system named WuKong that can surpass top e-sport players in *Honor of Kings* with a full hero pool. Further, Wu [37] and Gao *et al.* [8] proposed learning systems that enable the agent to learn human strategies to achieve policy diversity. However, these AI systems can only defeat human players but cannot collaborate well due to the communication gap between agents and humans, see Table 1. In most real-world scenarios, the excellent collaboration between humans and agents may make more sense than the competition.

## 2.2 Human-AI Collaboration

PBT is considered one way to solve the HAC problem [4]. Most PBT-based methods are devoted to training an agent which can be compatible with unseen partners by maintaining a population of agents with diverse behaviors in different ways [1, 2, 10, 11, 31, 41][6, 19, 20, 30]. These methods have been validated on both objective and subjective metrics in video games *Overcooked* and *Capture the Flag* and card game *Hanabi*. However, the main difference between these games and MOBA games is that these games do not provide explicit communication mechanics for collaboration on macro-strategies between agents and humans. Besides, MOBA AI agents usually need to learn billions of network parameters to cope with the enormous state-action spaces ($10^{20000}$) [38], which constitutes a prohibitive computational burden for learning. As a more realistic topic of HAC, human-robot interaction in manufacturing also attracts much attention [13, 17, 25]. However, these studies are mainly limited to collaboration between a robot and a human through one-way communication, i.e., humans give robots orders. Therefore, there is still a large room to study RL with the participation of humans. This work can be a stepping stone for broader real-world applications.

## 2.3 Multi-Agent Communication

Communication is often used in Multi-Agent Reinforcement Learning (MARL) to improve inter-agent collaboration. Most communication-based MARL methods are mainly focused on exploring communication protocols between multiple agents with an end-to-end RL framework [5, 7, 9, 23, 29, 32, 36]. Jiang and Lu [12] and Kim *et al.* [15] proposed to model the value of multi-agent communication for effective collaboration. Unfortunately, these methods all model communications in a latent space without considering human-AI interactions, making it less interpretable to humans. Instead, we focus on enabling humans and agents to collaborate through explicit communications.

## 3 Human-AI Collaboration

We consider an interpretable communicative human-AI collaboration task, which can be extended from Partially Observable Markov Decision Process (POMDP) and formulated as a tuple $< N, H, \mathbf{S}, \mathbf{A}^N, \mathbf{A}^H, \mathbf{O}, \mathbf{M}, r, P, \gamma >$, where $N$ and $H$ represent the numbers of agents and humans, respectively. $\mathbf{S}$ is the space of global states. $\mathbf{A}^N = \{A_i^N\}_{i=1,...,N}$ and $\mathbf{A}^H = \{A_i^H\}_{i=1,...,H}$ denote the spaces of actions of $N$ agents and $H$ humans, respectively. $\mathbf{O} = \{O_i\}_{i=1,...,N+H}$ denotes the space of observations of $N$ agents and $H$ humans. $\mathbf{M}$ represents the space of interpretable messages, that is, the Meta-Commands in the MCC framework. $P : \mathbf{S} \times \mathbf{A}^N \times \mathbf{A}^H \to \mathbf{S}$ and $r : \mathbf{S} \times \mathbf{A}^N \times \mathbf{A}^H \to \mathbb{R}$ denote the shared state transition probability function and reward function of $N$ agents, respectively. Note that, $r$ includes both individual reward and team reward. $\gamma \in [0, 1)$ denotes the discount factor. For each agent $i$ in state $s_t \in \mathbf{S}$, it receives an observation $o_t^i \in O_i$ and a selected message $c_t^i \in \mathbf{M}$, and then outputs an action $a_t^i = \pi_\theta(o_t^i, c_t^i) \in A_i^N$ and a new message $m_{t+1}^i = \pi_\phi(o_t^i) \in \mathbf{M}$, where $\pi_\theta$ and $\pi_\phi$ are action network and message encoding network, respectively. A message selector $c_t^i = \pi_\omega(o_t^i, C_t)$ is introduced to receive a message set $C_t = \{m_t^i\}_{i=1,...,N+H} \subset \mathbf{M}$ from all agents and humans and select the optimal one to execute.

We divide the HAC problem in MOBA games into the Human-to-AI (H2A) and the AI-to-Human (A2H) scenarios. The **H2A Scenario:** Humans send macro-strategies as messages to agent teammates, and agents combine them with their own messages to select the optimal one based on their own message selector to execute, achieving effective collaboration with humans. The **A2H Scenario:** Agents send messages as macro-strategies to human teammates, and humans combine them with their own macro-strategies to select the optimal one based on their own value systems to execute, achieving effective collaboration with agents. The goal of both tasks is that agents and humans communicate macro-strategies with pre-defined communication protocols, and then select valuable macro-strategies for effective collaboration to win the game.

## 4 Meta-Command Communication-Based Framework

In this section, we present the proposed MCC framework in detail. We first briefly describe three key stages of the MCC framework (see Section 4.1). Then we introduce the two pivotal modules in the MCC framework: 1) an interpretable communication protocol, i.e., the Meta-Command, as a general representation of macro-strategies to bridge the communication gap between agents and humans (see Section 4.2); 2) a meta-command value estimation model, i.e., the Meta-Command Selector, to select a valuable meta-command for each agent to achieve effective HAC in MOBA games(see Section 4.3).

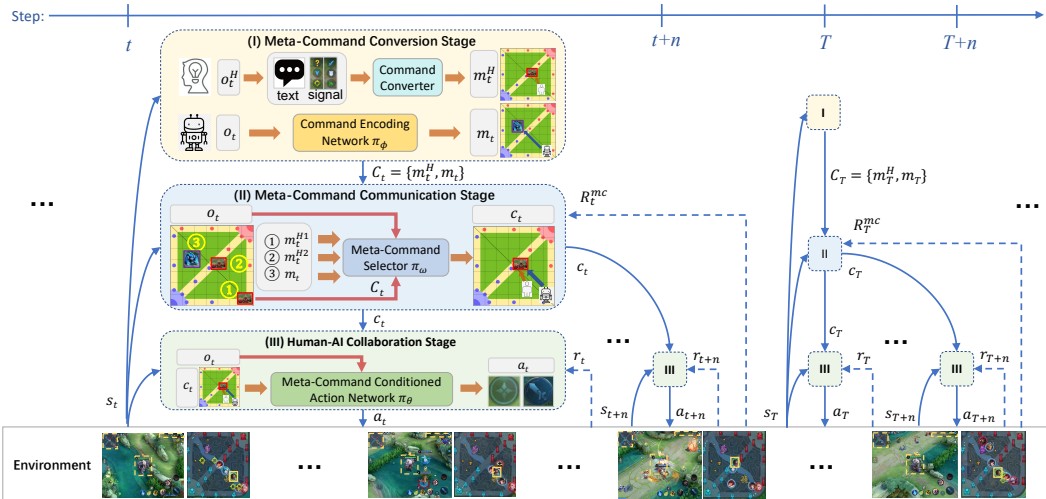

Figure 2: **The temporal process of the MCC framework.** For each communication step ($t$ and $T$), MCC first (I) converts messages from humans and agents into meta-commands, then (II) selects the optimal meta-command for each agent to execute, and (III) finally predicts a sequence of actions for each agent to perform. The selected meta-command is retained and executed for $n$ time steps. This process is repeated until the end of a game.

## 4.1 Overview

The flow of the MCC framework can be divided into three stages: the meta-command conversion stage, the meta-command communication stage, and the human-AI collaboration stage, as plotted in Figure 2. At the **Meta-Command Conversion Stage**, the MCC framework converts the macro-strategies sent by humans and the messages outputted by the command encoding network of agents into unified meta-commands and then broadcasts them to all agents and humans. At the **Meta-Command Communication Stage**, the MCC framework uses the meta-command selector to estimate the values of all received meta-commands and select the optimal one for each agent to execute. Note that humans also select the optimal meta-command based on their value systems. At the **Human-AI Collaboration Stage**, the MCC framework adopts the meta-command conditioned action network to predict a sequence of actions for each agent to perform based on its selected meta-command. For each game, humans and agents have to collaborate multiple times, that is, they need to perform the above three stages multiple times to win the game.

## 4.2 Meta-Command

In MOBA games, we propose that a macro-strategy consists of three components: where to go, what to do, and how long. For example, a macro-strategy can be *Come And Kill The Dragon*, which consists of *Come To The Dragon* (where to go), *Attack The Dragon* (what to do), and *Until The Dragon Is Killed* (how long). Thus, we propose a general representation of macro-strategies, i.e., the Meta-Command, as an interpretable communication protocol to bridge the communication gap between agents and humans.

**Meta-Command Definition.** We formulate the Meta-Command as a tuple $< L, E, T^{mc} >$, as shown in Figure 1(b), where $L$ is the *Location* to go, $E$ is the *Event* to do after reaching $L$, and $T^{mc}$ is the *Time Limit* for executing the meta-command. Among them, $L$ is the key to the meta-command, which contains the intention of the macro-strategy. $E$ can be thought of as human micro-operation, which is implemented through a pre-trained micro-action network $\pi_\theta$ in the MCC framework. $T^{mc}$ can be set to how long it normally takes a human to complete a macro-strategy in MOBA games, usually 20 seconds corresponds to 80% completion rate for meta-commands, see Appendix A.12.1.

**Meta-Command Conversion.** To realize interpretable human-AI communication, we convert the explicit messages from humans and the implicit messages from agents into unified meta-commands. To achieve the former, a hand-crafted command converter function $f^{cc}$ is used to generate $L$ of meta-commands by extracting the location from explicit messages, such as text and signals, sent by humans. To achieve the latter, we use a Command Encoding Network (CEN) $\pi_\phi(m|o)$ to generate $L$ of meta-commands. The CEN is trained via supervised learning (SL) with the goal of learning the distribution of meta-commands sent from humans, as shown in Figure 3(a)(I). The training dataset $\{< o, m >\}$ is obtained by extracting the observation $o$ and its corresponding meta-command $m$ from expert data.

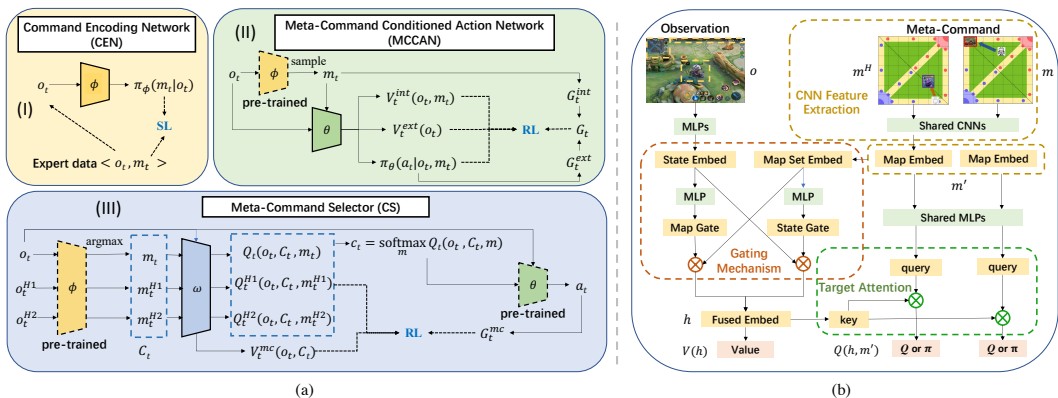

Figure 3: **The training process and model structure of MCC.** (a) The training process is divided into three phases: we first (I) train the CEN via supervised learning (SL), then (II) train the MCCAN via goal-conditioned RL, and finally (III) train the CS via RL. Among them, the dashed box represents the frozen model. (b) The detailed CS model structure, including CNN feature extraction, gating mechanism, target attention module, etc.

After converting all messages into unified meta-commands, the MCC framework broadcasts them to all agents and humans. Then, agents and humans receive an identical meta-command candidate set.

**Meta-Command Execution.** After receiving a meta-command candidate set, agents can select one meta-command from it to execute. We adopt a Meta-Command Conditioned Action Network (MCCAN) $\pi_\theta(a|o, m)$ for agents to perform actions based on the selected meta-command, as shown in Figure 3(a)(II). The MCCAN is trained via goal-conditioned RL with the goal of achieving a near-human completion rate for the meta-commands generated by the pre-trained CEN while ensuring that the win rate is not reduced. We adopt an intrinsic reward $r_t^{int}(s_t, m_t, s_{t+1}) = |f^{ce}(s_t) - m_t| - |f^{ce}(s_{t+1}) - m_t|$ to guide the process of executing the meta-command $m_t$, where $f^{ce}$ is a hand-crafted command extraction function. We train the MCCAN with the objective of maximizing the expectation over extrinsic and intrinsic discounted total rewards $G_t = \mathbb{E}_{s \sim d_{\pi_\theta}, a \sim \pi_\theta} \left[ \sum_{i=0}^{\infty} \gamma^i r_{t+i} + \alpha \sum_{j=0}^{T^{mc}} \gamma^j r_{t+j}^{int} \right]$, where $\alpha$ is a trade-off parameter and $d_\pi(s) = \lim_{t \to \infty} P \left( s_t = s \mid s_0, \pi \right)$ is the probability when following $\pi$ for $t$ steps from $s_0$.

After training the CEN and MCCAN, we can achieve HAC by simply setting an agent to randomly select a meta-command derived from humans to execute. However, such collaboration is non-intelligent and can even be a disaster for game victory because agents have no mechanism to model the values of meta-commands and cannot choose the optimal meta-command to execute. While humans usually choose the optimal one based on their value systems for achieving effective collaboration to win the game. Thus, we further propose a meta-command value estimation model to select a valuable meta-command for each agent, as described in the following subsection.

### 4.3 Meta-Command Selector

In real-world MOBA games, the same macro-strategy often has different values for different humans in different situations. For example, a macro-strategy can be *Come And Kill The Dragon*, as shown in Figure 1(b). It is more valuable for humans A and B to collaborate. While another macro-strategy can be *Clean Up Top-Lane Minions*, which is more valuable for human C rather than humans A and B. Therefore, it is important to select the most valuable meta-command from the received meta-command candidate set $C$ to achieve effective human-AI collaboration. We propose a meta-command value estimation model, i.e., the Meta-Command Selector (CS) $\pi_\omega(o, C)$, to estimate the values of all current meta-commands and select the most valuable one for each agent to execute.

**CS Optimization Objective.** Typically, the execution of a meta-command involves reaching location $L$ and doing event $E$, of which the latter is more important to the value of the meta-command. For example, for the meta-command *Come And Kill The Dragon*, if *Kill The Dragon* event cannot be done within $T^{mc}$ time steps, then it is pointless to *Come To The Dragon*. Thus, the long-term reward $R^{mc}$ for executing a meta-command can be expressed as the total rewards within $T^{mc}$ time steps by interacting with the environment: $R_t^{mc} = \sum_{i=0}^{T^L} r_{t+i} + \beta \sum_{j=T^L}^{T^{mc}} r_{t+j}$, where $T^L < T^{mc}$ is the time for reaching $L$ and $\beta > 1$ is a trade-off parameter. Note that the reward function $r$

includes both individual rewards and team rewards. The optimization objective of CS is to select the optimal meta-command $m_t^* = \pi_\omega(o_t, C_t)$ for each agent to maximize the expected discounted meta-command execution return $G_t^{mc} = \mathbb{E}_{s \sim d_{\pi_\theta}, m \sim \pi_\omega, a \sim \pi_\theta} \left[ \sum_{i=0}^{\infty} \gamma_{mc}^i R_{t+i \cdot T^{mc}}^{mc} \right]$, where $o_t \in \mathbf{O}$, $C_t$ is the meta-command candidate set in state $s_t$, and $\gamma_{mc} \in [0, 1)$ is the discount factor.

**CS Training Process.** We construct a self-play training environment for CS where agents can send messages to each other. Specifically, three tricks in Figure 3(a)(III) are adopted to increase the sample efficiency while ensuring efficient exploration. First, each sent meta-command $m$ is sampled with the argmax rule from the results predicted by the pre-trained CEN. Second, each agent sends its meta-command with a probability $p$ every $T^{mc}$ time steps. Finally, each agent selects the final meta-command $c$ sampled with the softmax rule from its CS output results and hands it over to the pre-trained MCCAN for execution. We use the multi-head value mechanism [38] to model the value of the meta-command execution, and the corresponding value loss can be formulated as:

$$L^V(\omega) = \mathbb{E}_{S,C} \left[ \sum_{head_k} \|G_k^{mc} - V_\omega^k(S, C)\|_2 \right],$$

where $V_\omega^k(S, C)$ is the value of the $k$-th head. For DQN-based methods [21, 33, 35], the $Q$ loss is:

$$L^Q(\omega) = \mathbb{E}_{S,C,M} \left[ \|G_{total} - Q_\omega^k(S, C, M)\|_2 \right], G_{total} = \sum_{head_k} w_k G_k^{mc},$$

where $w_k$ is the weight of the $k$-th head and $G_k^{mc}$ is the Temporal Difference (TD) estimated value error $R_k^{mc} + \gamma_{mc} V_\omega^k(S', C') - V_\omega^k(S, C)$.

**CS Model Structure.** We design a general network structure for CS towards MOBA games, as shown in Figure 3(b). In MOBA games, the meta-commands corresponding to adjacent regions usually have similar values. Thus, we divide the meta-commands in the map into grids, a common location description for MOBA games, and use the shared Convolutional Neural Network (CNN) to extract region-related information from the meta-commands to improve the generalization of CS to adjacent meta-commands. Besides, we use the gating mechanism [18] to fuse the map embedding of all received meta-commands and the state embedding of the observation information. Finally, to directly construct the relationship between the observation information and each meta-command, we introduce a target attention module, where the query is the fused embedding $h$ and the key is the map embedding $m'$ of each meta-command. The fused embedding $h$ is used as the input into the subsequent Q network $Q(h, m')$ and V network $V(h)$ network of CS. In this way, the Q network can also be easily converted to the policy network $\pi(m|h, m')$. Thus, the CS model structure can be easily applied to most popular RL algorithms, such as PPO [24], DQN [21], etc.

# 5 Experiments

We evaluate the proposed MCC framework in *Honor of Kings*, one of the most popular MOBA games worldwide, which has been actively used as the testbed for recent game AI research [8, 37–40]. We conduct all experiments in *Honor of Kings* 5v5 mode with a full hero pool (over 100 heroes), except ablation studies with a 20 hero pool for exploring the influence of different model components more sufficiently and efficiently.

## 5.1 Experimental Setup

### 5.1.1 Training Setup [1]

Due to the complexity of MOBA games and limited resources, instead of training jointly, we train the CEN, MCCAN, and CS sequentially. For all model training, the location $L$ of meta-commands in the map is divided into 144 grids. The time limit $T^{mc}$ for the meta-command execution is set to 20s.

**CEN Training Settings.** We train the CEN via SL until it converges for 26 hours using 8 NVIDIA P40 GPUs. The batch size of each GPU is set to 512. Adam[16] is adopted as the optimizer with an initial learning rate of 0.0001.

**MCCAN Training Settings.** We train the MCCAN by finetuning a pre-trained micro-action network [38], the state-of-the-art (SOTA) model in *Honor of Kings*, which is conditioned on the meta-command sampled from the pre-trained CEN. The MCCAN is trained until it converges for 48

---

[1]Detailed parameter settings for all training processes can be found in the Appendix.

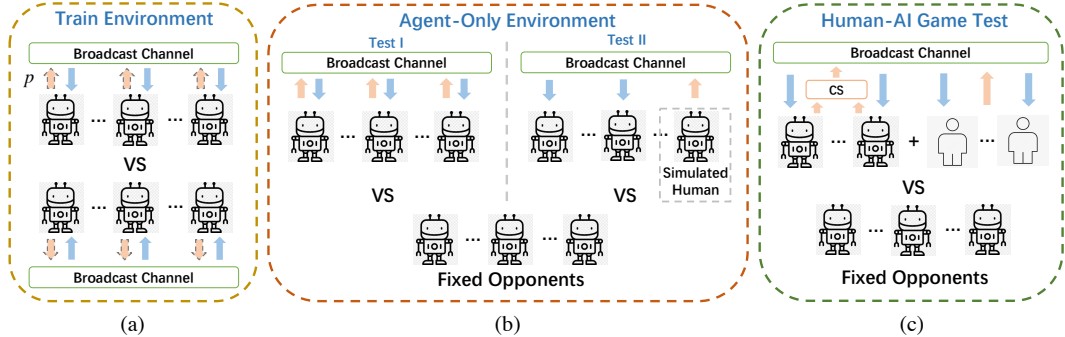

(a)               (b)               (c)

Figure 4: **Communication environments in the experiment.** The orange arrows indicate sending meta-commands, and the blue arrows indicate receiving meta-commands. The dashed line denotes sending meta-commands with probability $p$.

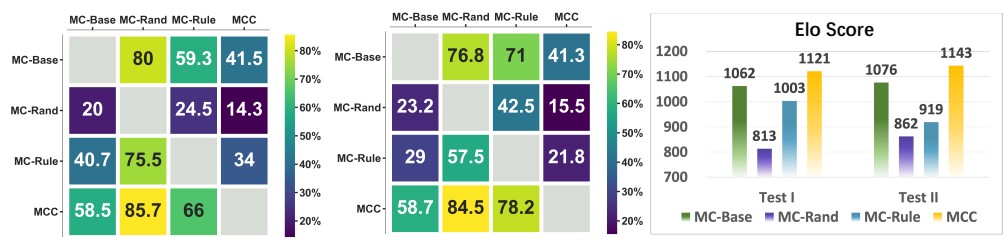

(a) Win rate (row vs. col) in Test I.    (b) Win rate (row vs. col) in Test II.    (c) Elo scores of different agents.

Figure 5: **AI performance in the testing environments.** (a) and (b) show the win rate maps of different agents who play against each other. (c) shows the final Elo scores of these agents.

hours using a physical computer cluster with 63,000 CPUs and 560 NVIDIA V100 GPUs. The batch size of each GPU is set to 256.

**CS Training Settings.** We train the CS via self-play until it converges for 24 hours using a physical computer cluster with 70,000 CPUs and 680 NVIDIA V100 GPUs. The batch size of each GPU is set to 256. The parameter $\beta$ is set to 2. Each agent sends a meta-command with a probability $p$ of 0.8 and an interval $T^{mc}$ of 20s, as shown in Figure 4(a).

### 5.1.2 Evaluating Setup

Our primary concern is whether the agents trained with the MCC framework, briefly called the MCC agents, can collaborate with humans well. However, evaluating agents with humans is expensive, which is not conducive to model selection and iteration. Therefore, we built two agent-only testing environments: Test I and Test II, for the model selection and iteration process, as shown in Figure 4(b). We also evaluate the MCC agents in practical human-AI game tests to examine the performance of collaborating with humans, as shown in Figure 4(c).

**Compared Agents.** We compare the MCC agent with three different types of agents: the MC-Base agent (agent only executes its own meta-command without communication), the MC-Rand agent (agent randomly selects a meta-command to execute), and the MC-Rule agent (agent selects the nearest meta-command to execute). We adopt the MC-Base agent-only team as the opponent for all tests. Note that the MC-Base agent-only team has the ability of the SOTA and is more stable than the human-only team. Results are reported over five random seeds.

**Agent-Only Environmental Settings.** Test I is the most complex environment where all agent teammates can send and receive meta-commands simultaneously with an interval of 20s. Test I is used to evaluate the agents' performance under extremely complex situations as well as in ablation studies. Test II is a simple environment to simulate practical game scenarios, where at most one human sends his macro-strategy at a time step. Thus, in Test II, only one agent is randomly selected to send its meta-command with an interval of 20s, and the other agents only receive meta-commands.

**Human-AI Game Testing Settings.** We had different types of agents team up with different levels and numbers of humans, including 15 strong humans (top1%) and 15 average humans (top30%), in *m AI + n Human* mode, where $m + n = 5$. For fair comparisons, each tester was not told the

type of agent teammates. To eliminate the effects of collaboration between agents, we prohibit agents from receiving meta-commands from their agent teammates, and the agent can only receive meta-commands from humans. In each game test, humans can send the converted meta-commands whenever they think their macro-strategies are important. To make the agent behave like humans (at most one human sends his macro-strategy at a time step), we restrict agents from sending their meta-commands. We randomly choose a human teammate and use his observation and all agents' meta-commands as the CS input and select the final output of CS to send with an interval of 20s.

## 5.2 Results in Agent-Only Environment

### 5.2.1 AI Performance

The Kullback-Leibler (KL) divergence of the meta-command distribution between the CEN and humans decreased from 4.96 to 0.44 as training converges. The MCCAN is trained with the parameter $\alpha$ equal to 16. The win rate of the trained agent against the SOTA agent [8, 38] is close to 50%. The average completion rates of the trained agent and humans for meta-commands are 82% and 80%, respectively. Notably, we can train an agent with a higher completion rate by increasing $\alpha$, but this will significantly reduce the win rate because the meta-command executed is not necessarily optimal and may result in the death of agents. We put the detailed experimental results of the CEN and MCCAN in the Appendix A.10.1 and A.10.2 due to space limitations.

Figure 5(a) and (b) show the win rates of four types of agents who play against each other for 600 matches in Test I and Test II, respectively. We see that the MCC agent achieves the highest win rate against all the other agents in both testing environments, indicating that the CS can select a valuable meta-command for each agent to collaborate, and such reasonable collaboration is conducive to winning the game. The MC-Rand and MC-Rule agents are worse than the MC-Base agent, confirming that agents executing low-value meta-commands can hurt performance. Notably, we find that the win rates of the MCC agent in Test I and Test II are close, suggesting that the MCC agent can generalize to different numbers of meta-commands. Figure 5(c) demonstrates the final Elo scores [3] of these agents. It clearly shows the effectiveness of CS in agent-only collaboration scenarios.

### 5.2.2 Ablation Studies

We further investigate the influence of different components, including CNN feature extraction with the gating mechanism (w/o CNN-GM), target attention module (w/o TA), and PPO optimization algorithm (MCC-PPO), on the performance of CS. We conduct ablation studies in Test I with a 20 hero pool. In practical games, meta-commands

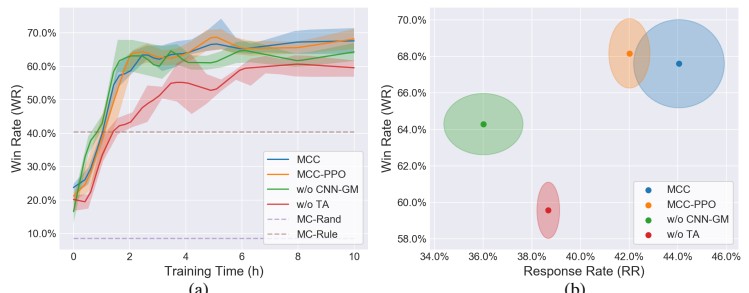

Figure 6: **Results of ablation studies.** (a) The training curves of different CS ablation versions. (b) The converged WR-RR results of different CS ablation versions. The shadow indicates the standard deviation.

with adjacent regions often have similar intentions and values. Thus the response rate of the agent to adjacent meta-commands should be as close as possible. Besides, the higher the agent's response rate to meta-commands, the more collaborative behaviors of the agent, thus we expect the response rate of CS as high as possible. Generally, we expect the Response Rate (RR) of CS as high as possible while ensuring that the Win Rate (WR) is not reduced.

Figure 6(a) demonstrates the WR of different CS ablation versions during the training process, and Figure 6(b) shows the converged WR-RR results. We see that after ablating the TA module, the WR and RR of CS are greatly reduced, indicating that the TA module can improve the accuracy of CS to meta-commands. Besides, after ablating the CNN-GM module, the RR of CS is most affected, which is reduced by 20%. It indicates that without the CNN-GM module, the value estimation of CS to adjacent meta-commands is not accurate enough, resulting in missing some actual high valuable meta-commands. We notice that the MCC and MCC-PPO in both metrics are close, confirming the versatility of the CS model structure.

Table 1: The WR of different human-AI teams against MC-Base agents in *4 AI + 1 Human* mode.

| Teammate | Type of Agent | | |
|---|---|---|---|
| | MC-Base | MC-Rand | MCC |
| Average Human | 23% | 5% | **37%** |
| Strong Human | 42% | 28% | **54%** |

Table 2: The RR of humans and agents to teammates.

| Sender\Receiver | Average Human | Strong Human | MCC |
|---|---|---|---|
| MC-Rand | 41.07% | 35.69% | 34.03% |
| Average Human | 72.34% | - | 61.17% |
| Strong Human | - | **74.91%** | **73.05%** |
| MCC | **73.43%** | **78.50%** | - |

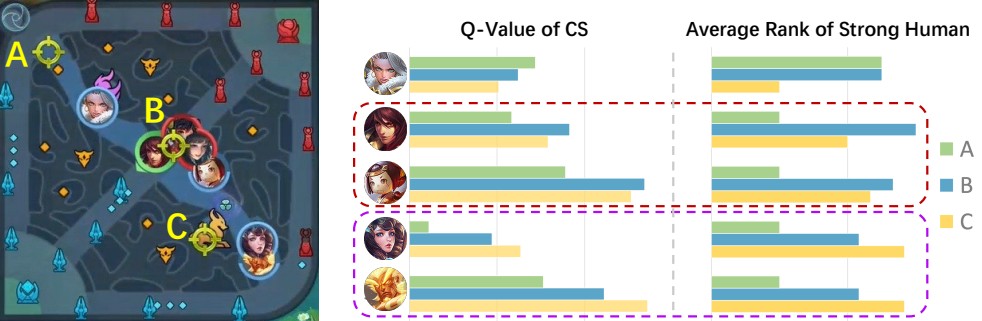

Figure 7: **Case study on the value estimation of CS.**

## 5.3 Results in Human-AI Game Test

Due to space limitations, we only show the objective results in *4 AI + 1 Human* mode. Other modes results and the subjective preference results of testers can be found in the Appendix A.10.3 and A.11. Table 1 shows the WR of different human-AI teams who play against the MC-Base agent-only team. We see that the MCC agent significantly outperforms other agents, regardless of whether they pair with a strong or average human. To explain why humans have a higher WR when paired with the MCC agents, we count the RR of agents to the meta-commands sent from human teammates (H2A scenarios) and the RR of humans to the meta-commands sent from agent teammates (A2H scenarios), respectively, as shown in Table 2. In H2A scenarios, the RRs of the MCC agents to average humans and strong humans are 61.17% and 73.05%, respectively, indicating that the MCC agents are more willing to respond to valuable meta-commands sent from strong humans. We also notice that the RR of the MCC agents to strong humans (73.05%) is very close to the RR of strong humans themselves (74.91%), suggesting that the CS is close to the value system of strong humans. In A2H scenarios, the RRs of average humans and strong humans to the MCC agents are 73.43% and 78.5%, respectively, which is significantly higher than that of MC-Rand agents (41.07% and 35.69%), indicating that the meta-commands sent from the MCC agents are more valuable and reasonable to humans. Note that the RR of the MCC agents to the MC-Rand agents is 34.03%, which is close to that of strong humans (35.69%), once again confirming that the CS is close to the value system of strong humans.

We also visualize the comparison of CS and strong human value systems on a game scene with three meta-commands existing, as shown in Figure 7. We see that the CS selects the meta-command B for the two heroes in the red dashed box to collaborate, selects the meta-command C for the two heroes in the purple dashed box to collaborate, and selects the meta-command A for the remaining hero to execute alone. The CS selection results are consistent with the ranking results of strong humans, confirming the effectiveness of CS and the interpretability of the collaboration behavior between MCC agents and humans.

## 6 Conclusion

In this paper, we proposed an efficient and interpretable Meta-Command Communication-based framework, dubbed MCC, to achieve effective human-AI collaboration in MOBA games. To bridge the communication gap between humans and agents, we designed an interpretable communication protocol, i.e., the Meta-Command, to convert the explicit messages from humans and the implicit messages from agents into unified meta-commands. To achieve effective collaboration, we constructed a meta-command value estimation model, i.e., the Meta-Command Selector, to select a valuable meta-command for each agent to execute. Finally, we introduced the training process of the MCC framework and conducted practical human-AI game tests in the typical MOBA game *Honor of Kings*. The experimental results show that the MCC agents can collaborate reasonably with human teammates and even generalize to collaborate with different levels and numbers of human teammates. We expect this work can be a foundation for future HAC research in complex environments.

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
