# A  Appendix

## A.1  Game Environment

Figure 1 shows the UI interface of *Honor of Kings*. For fair comparisons, all experiments in this paper were carried out using a fixed released gamecore version (Version 3.73 series) of *Honor of Kings*.

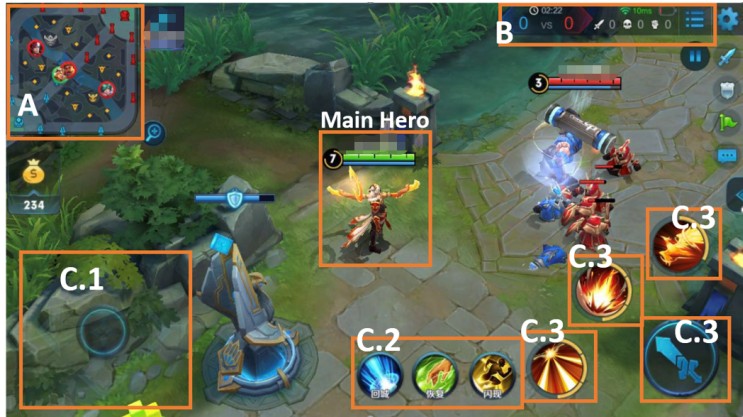

Figure 1: **The UI interface of *Honor of Kings*.** The hero controlled by the player is called *Main Hero*. The player controls the hero's movement through the bottom-left wheel (C.1) and releases the hero's skills through the bottom-right buttons (C.2, C.3). The player can observe the local view via the screen, observe the global view via the top-left mini-map (A), and obtain game states via the top-right dashboard (B).

## A.2  In-game Signaling System

Figure 2 demonstrates the in-game signaling system of *Honor of Kings*. Players can communicate and collaborate with teammates through the in-game signaling system. In the **Human-AI Game Test**, humans can send macro-strategies to agents through signals like A in figure 2, and these signals are displayed to teammates in the form of D. The MCC framework converts these explicit messages, i.e., signals, into meta-commands by the hand-crafted command converter function $f^{cc}$ and broadcast them to all agent teammates. And the MCC framework can also convert the meta-commands sent from agents into signals by the inverse of $f^{cc}$ and broadcast them to all human teammates.

Voice (B.2) and text (B.1 and B.3) are two other forms of communication. In the future, we consider introducing a general meta-command encoding model that can handle all forms of explicit messages (signals, voice, and text).

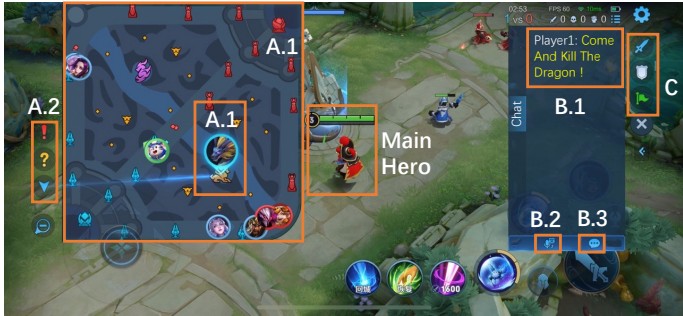 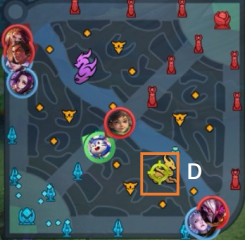

Figure 2: **The in-game signaling system of *Honor of Kings*.** Players can send their macro-strategies by dragging signal buttons (A.2) to the corresponding locations (A.1) in the mini-map. The sent result is displayed in the form of a yellow circle (D). C is the convenience signals representing attack, retreat, and assembly, respectively. Voice (B.2) and text (B.1 and B.3) are two other forms of communication.

## A.3 Hero Pool

Table 1 shows the full hero pool and 20 hero pool used in the **Experiments**. Each match involves two lineups playing against each other, and each lineup consists of five randomly picked heroes.

Table 1: Hero pool used in the **Experiments**.

| | |
|---|---|
| Full Hero pool | Lian Po, Xiao Qiao, Zhao Yun, Mo Zi, Da Ji, Ying Zheng, Sun Shangxiang, Luban Qihao, Zhuang Zhou, Liu Chan Gao Jianli, A Ke, Zhong Wuyan, Sun Bin, Bian Que, Bai Qi, Mi Yue, Lv Bu, Zhou Yu, Yuan Ge Xia Houdun, Zhen Ji, Cao Cao, Dian Wei, Gongben Wucang, Li Bai, Make Boluo, Di Renjie, Da Mo, Xiang Yu Wu Zetian, Si Mayi, Lao Fuzi, Guan Yu, Diao Chan, An Qila, Cheng Yaojin, Lu Na, Jiang Ziya, Liu Bang Han Xin, Wang Zhaojun, Lan Lingwang, Hua Mulan, Ai Lin, Zhang Liang, Buzhi Huowu, Nake Lulu, Ju Youjing, Ya Se Sun Wukong, Niu Mo, Hou Yi, Liu Bei, Zhang Fei, Li Yuanfang, Yu Ji, Zhong Kui, Yang Yuhuan, Chengji Sihan Yang Jian, Nv Wa, Ne Zha, Ganjiang Moye, Ya Dianna, Cai Wenji, Taiyi Zhenren, Donghuang Taiyi, Gui Guzi, Zhu Geliang Da Qiao, Huang Zhong, Kai, Su Lie, Baili Xuance, Baili Shouyue, Yi Xing, Meng Qi, Gong Sunli, Shen Mengxi Ming Shiyin, Pei Qinhu, Kuang Tie, Mi Laidi, Yao, Yun Zhongjun, Li Xin, Jia Luo, Dun Shan, Sun Ce Zhu Bajie, Shangguan Waner, Ma Chao, Dong Fangyao, Xi Shi, Meng Ya, Luban Dashi, Pan Gu, Chang E, Meng Tian Jing, A Guduo, Xia Luote, Lan, Sikong Zhen, Erin, Yun ying, Jin Chan, Fei, Sang Qi |
| 20 Hero Pool | Jing, Pan Gu, Zhao Yun, Ju Youjing, Donghuang Taiyi, Zhang Fei, Gui Guzi, Da Qiao, Sun Shangxiang, Luban Qihao, Chengji Sihan, Huang Zhong, Zhuang Zhou, Lian Po, Liu Bang, Zhong Wuyan, Yi Xing, Zhou Yu, Xi Shi, Zhang Liang |

## A.4 Agent Action

Table 2 shows the action space of agents.

Table 2: The action space of agents.

| Action | Detail | Description |
|---|---|---|
| | Illegal action | Placeholder. |
| | None action | Executing nothing or stopping continuous action. |
| | Move | Moving to a certain direction determined by move x and move y. |
| | Normal Attack | Executing normal attack to an enemy unit. |
| | Skill1 | Executing the first skill. |
| | Skill2 | Executing the second skill. |
| What | Skill3 | Executing the third skill. |
| | Skill4 | Executing the fourth skill (only a few heroes have Skill4). |
| | Summoner ability | An additional skill choosing before the game begins (10 to choose). |
| | Return home(Recall) | Returning to spring, should be continuously executed. |
| | Item skill | Some items can enable an additional skill to player's hero. |
| | Restore | Blood recovering continuously in 10s, can be disturbed. |
| | Collaborative skill | Skill given by special ally heroes. |
| | Move X | The x-axis offset of moving direction. |
| How | Move Y | The y-axis offset of moving direction. |
| | Skill X | The x-axis offset of a skill. |
| | Skill Y | The y-axis offset of a skill. |
| Who | Target unit | The game unit(s) chosen to attack. |

## A.5 Reward Design

Table 3 demonstrates the details of the designed environment reward.

## A.6 Infrastructure Design

Figure 3 shows the infrastructure of the training system, which consists of four pivotal components: AI Server, Inference Server, RL Learner, and Memory Pool. The AI Server (the actor) covers the interaction logic between the agents and the environment. The Inference Server is used for the centralized batch inference on the GPU side. The RL Learner (the learner) is a distributed training environment for RL models. And the Memory Pool is used for storing the experience, implemented as a memory-efficient circular queue.

As is known to all, training complex game AI systems often require a large amount of computing resources, such as AlphaGo Lee Sedol (280 GPUs), OpenAI Five Final (1920 GPUs), and AlphaStar Final (3072 TPUv3 cores), we also use hundreds of GPUs for training the agents. Another future work is to improve resource utilization using fewer computing resources.

Table 3: The details of the environment reward.

| Head | Reward Item | Weight | Type | Description |
|------|-------------|--------|------|-------------|
| Farming Related | Gold | 0.005 | Dense | The gold gained. |
| | Experience | 0.001 | Dense | The experience gained. |
| | Mana | 0.05 | Dense | The rate of mana (to the fourth power). |
| | No-op | -0.00001 | Dense | Stop and do nothing. |
| | Attack monster | 0.1 | Sparse | Attack monster. |
| KDA Related | Kill | 1 | Sparse | Kill a enemy hero. |
| | Death | -1 | Sparse | Being killed. |
| | Assist | 1 | Sparse | Assists. |
| | Tyrant buff | 1 | Sparse | Get buff of killing tyrant, dark tyrant, storm tyrant. |
| | Overlord buff | 1.5 | Sparse | Get buff of killing the overlord. |
| | Expose invisible enemy | 0.3 | Sparse | Get visions of enemy heroes. |
| | Last hit | 0.2 | Sparse | Last hitting an enemy minion. |
| Damage Related | Health point | 3 | Dense | The health point of the hero (to the fourth power). |
| | Hurt to hero | 0.3 | Sparse | Attack enemy heroes. |
| Pushing Related | Attack turrets | 1 | Sparse | Attack turrets. |
| | Attack crystal | 1 | Sparse | Attack enemy home base. |
| Win/Lose Related | Destroy home base | 2.5 | Sparse | Destroy enemy home base. |

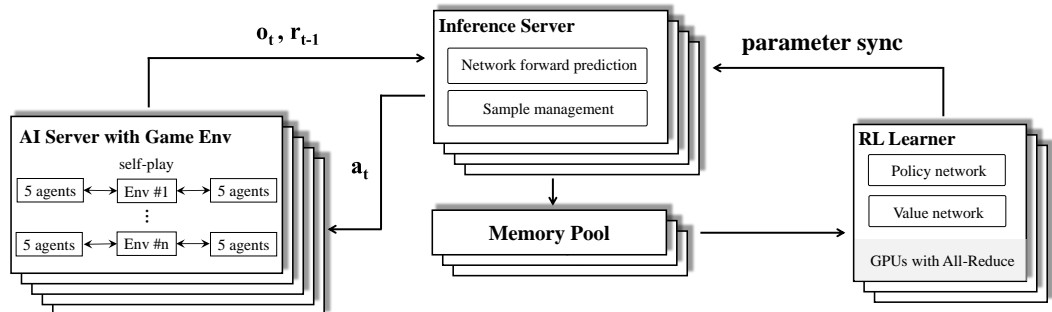

Figure 3: The designed infrastructure.

## A.7 Feature Design

### A.7.1 CEN

See Table 4.

### A.7.2 MCCAN

See Table 5.

### A.7.3 Feature of CS

See Table 6.

## A.8 Network Architecture

### A.8.1 CEN

Figure 4 shows the detailed model structure of CEN. The CEN predicts a meta-command Softmax distribution for each agent based on its current observation. The outputted meta-command indicates the macro-strategy for future $T^{mc}$ steps.

### A.8.2 MCCAN

Figure 5 shows the detailed model structure of MCCAN. The MCCAN predicts a sequence of actions for each agent based on its observation and the meta-command sampled from the Top-$k$ Softmax distribution of CEN. The observations are processed by a deep LSTM, which maintains memory

Table 4: Feature details of CEN.

| Feature Class | Field | Description | Dimension |
|---|---|---|---|
| **1. Unit feature** | Scalar | Includes heroes, minions, monsters, and turrets | 3946 |
| Heroes | Status | Current HP, mana, speed, level, gold, KDA, and magical attack and defense, etc. | 1562 |
|  | Position | Current 2D coordinates | 20 |
| Minions | Status | Current HP, speed, visibility, killing income, etc. | 920 |
|  | Position | Current 2D coordinates | 80 |
| Monsters | Status | Current HP, speed, visibility, killing income, etc. | 728 |
|  | Position | Current 2D coordinates | 56 |
| Turrets | Status | Current HP, locked targets, attack speed, etc. | 540 |
|  | Position | Current 2D coordinates | 40 |
| **2. In-game stats feature** | Scalar | Real-time statistics of the game | 104 |
| Static statistics | Time | Current game time | 57 |
|  | Camp | Types of two camps | 1 |
|  | Alive heroes | Number of alive heroes of two camps | 10 |
|  | Kill | Kill number of each camp | 6 |
|  | Alive turrets | Number of alive turrets of two camps | 8 |
| Comparative statistics | Alive heroes diff | Alive heroes difference between two camps | 11 |
|  | Kill diff | Kill difference between two camps | 5 |
|  | Alive turrets diff | Alive turrets difference between two camps | 6 |

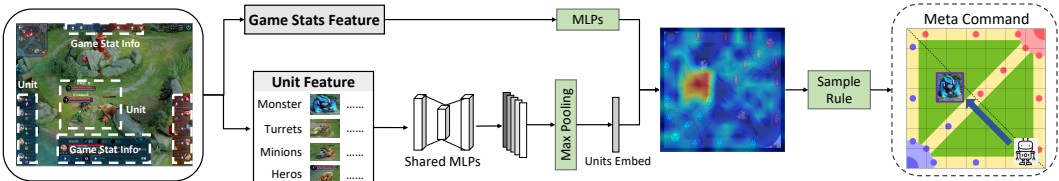

Figure 4: The CEN model structure.

among steps. we use the target attention mechanism to improve the accuracy of the model prediction, and we design the action mask module to eliminate unnecessary actions for efficient exploration. Additionally, we introduce a value mixer module [4] to model team value for improving the accuracy of the value estimation. Finally, following [5] and [1], we adopt hierarchical heads of actions, including three parts: 1) What action to take; 2) who to target; 3) how to act.

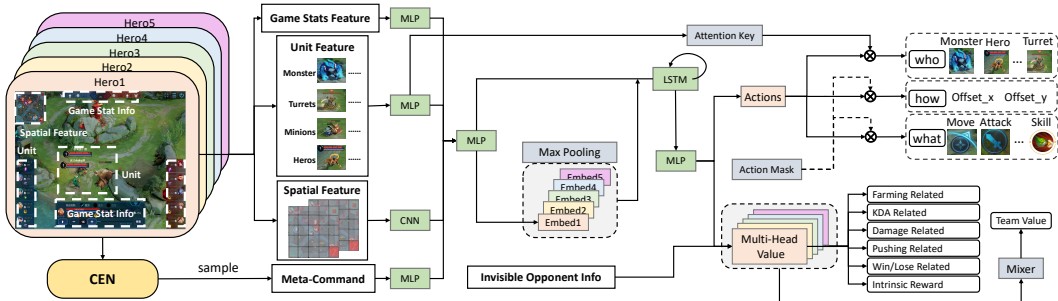

Figure 5: The MCCAN model structure.

## A.9 Details of Human-AI Game Test

### A.9.1 Participant

We contacted the game provider and got a test authorization. The game provider found us participants who meet the requirements. During the **Human-AI Game Test**, we only know the rank-level and game experience information of participants and do not know their identity information. And special equipment and game accounts are provided to the participants to prevent the leakage of equipment

Table 5: Feature details of MCCAN.

| Feature Class | Field | Description | Dimension |
|---|---|---|---|
| **1. Unit feature** | Scalar | Includes heroes, minions, monsters, and turrets | 8599 |
| Heroes | Status | Current HP, mana, speed, level, gold, KDA, buff, bad states, orientation, visibility, etc. | 1842 |
| | Position | Current 2D coordinates | 20 |
| | Attribute | Is main hero or not, hero ID, camp (team), job, physical attack and defense, magical attack and defense, etc. | 1330 |
| | Skills | Skill 1 to Skill N's cool down time, usability, level, range, buff effects, bad effects, etc. | 2095 |
| | Item | Current item lists | 60 |
| Minions | Status | Current HP, speed, visibility, killing income, etc. | 1160 |
| | Position | Current 2D coordinates | 80 |
| | Attribute | Camp (team) | 80 |
| | Type | Type of minions (melee creep, ranged creep, siege creep, super creep, etc.) | 200 |
| Monsters | Status | Current HP, speed, visibility, killing income, etc. | 868 |
| | Position | Current 2D coordinates | 56 |
| | Type | Type of monsters (normal, blue, red, tyrant, overlord, etc.) | 168 |
| Turrets | Status | Current HP, locked targets, attack speed, etc. | 520 |
| | Position | Current 2D coordinates | 40 |
| | Type | Type of turrets (tower, high tower, crystal, etc.) | 80 |
| **2. In-game stats feature** | Scalar | Real-time statistics of the game | 68 |
| Static statistics | Time | Current game time | 5 |
| | Gold | Golds of two camps | 12 |
| | Alive heroes | Number of alive heroes of two camps | 10 |
| | Kill | Kill number of each camp | 6 |
| | Alive turrets | Number of alive turrets of two camps | 8 |
| Comparative statistics | Gold diff | Gold difference between two camps | 5 |
| | Alive heroes diff | Alive heroes difference between two camps | 11 |
| | Kill diff | Kill difference between two camps | 5 |
| | Alive turrets diff | Alive turrets difference between two camps | 6 |
| **3. Invisible opponent information** | Scalar | Invisible information used for the value net | 560 |
| Opponent heroes | Position | Current 2D coordinates, distances, etc. | 120 |
| NPC | Position | Current 2D coordinates of all non-player characters, including minions, monsters, and turrets | 440 |
| **4. Spatial feature** | Spatial | 2D image-like, extracted in channels for convolution | 6x17x17 |
| Skills | Region | Potential damage regions of ally and enemy skills | 2x17x17 |
| | Bullet | Bullets of ally and enemy skills | 2x17x17 |
| Obstacles | Region | Forbidden region for heroes to move | 1x17x17 |
| Bushes | Region | Bush region for heroes to hide | 1x17x17 |
| **5. Meta-Command feature** | Spatial | Flattened Meta-Command | 144 |

and account information. The game statistics we collect are for experimental purposes only and are not disclosed to the public.

The participants consisted of 15 strong humans (top 1%) and 15 average humans (top 30%). All participants have more than three years of experience in *Honor of Kings* and promise to be familiar with all mechanics in the game, including the in-game signaling system in Figure 2. We used *m AI + n Human* mode to evaluate the performance of agents teaming up with different numbers of humans, where $m + n = 5$. Each participant is asked to randomly team up with three different types of agents, including the MC-Base agents, the MC-Rand agents, and the MCC agents. For fair comparisons, we adopt the MC-Base agent as the opponent for all tests. Each participant tested 20 matches for the *4 AI + 1 Human* mode. Each strong human participant tested additional 10 matches for the *3 AI + 2 Human* and *2 AI + 3 Human* modes, respectively. In all tests, participants were not told the type of agent teammates.

In addition, as mentioned in [5, 1], the response time of agents is usually set to 193ms, including observation delay (133ms) and response delay (60ms). The average APMs of agents and top e-sport players are usually comparable (80.5 and 80.3, respectively). To make our test results more accurate,

Table 6: Feature details of CS.

| Feature Class | Field | Description | Dimension |
|---|---|---|---|
| **1. Unit feature** | Scalar | Includes heroes, minions, monsters, and turrets | 3946 |
| Heroes | Status | Current HP, mana, speed, level, gold, KDA, and magical attack and defense, etc. | 1562 |
| | Position | Current 2D coordinates | 20 |
| Minions | Status | Current HP, speed, visibility, killing income, etc. | 920 |
| | Position | Current 2D coordinates | 80 |
| Monsters | Status | Current HP, speed, visibility, killing income, etc. | 728 |
| | Position | Current 2D coordinates | 56 |
| Turrets | Status | Current HP, locked targets, attack speed, etc. | 540 |
| | Position | Current 2D coordinates | 40 |
| **2. In-game stats feature** | Scalar | Real-time statistics of the game | 104 |
| Static statistics | Time | Current game time | 57 |
| | Camp | Types of two camps | 1 |
| | Alive heroes | Number of alive heroes of two camps | 10 |
| | Kill | Kill number of each camp | 6 |
| | Alive turrets | Number of alive turrets of two camps | 8 |
| Comparative statistics | Alive heroes diff | Alive heroes difference between two camps | 11 |
| | Kill diff | Kill difference between two camps | 5 |
| | Alive turrets diff | Alive turrets difference between two camps | 6 |
| **3. Invisible opponent information** | Scalar | Invisible information used for the value net | 560 |
| Opponent heroes | Position | Current 2D coordinates, distances, etc. | 120 |
| NPC | Position | Current 2D coordinates of all non-player characters, including minions, monsters, and turrets | 440 |
| **4. Meta-Command feature** | Spatial | 2D image-like, extracted in channels for convolution | 5x12x12 |
| Meta-Commands | Spatial | All received Meta-Commands in the team | 5x12x12 |

we adjusted the capability of agents to match the performance of strong humans by increasing the observation delay (from 133ms to 200ms) and response delay (from 60ms to 120 ms).

## A.9.2 Test Introduction

All participants were told the following instructions before testing:

- You will be invited into matches where your opponents and teammates are agents.
- Your goal is to win the game as much as possible by collaborating with agent teammates.
- You can collaborate with agent teammates through the in-game signaling system, just like playing with human teammates.
- In addition, agent teammates will also send you signals representing their macro-strategies, and you can judge whether to execute them based on your value system.
- Each game is about 10-20 minutes. Your identity information will not be disclosed to anyone, and all game statistics are only used for academic research. You will voluntarily choose whether to take the test.

If the participant volunteers to take the test, we will provide the equipment and game account to him, and the test will begin.

## A.9.3 Potential Participant Risks

First, we analyze the risks of this experiment to the participants. The potential participant risks of the experiment mainly include the leakage of identity information and the time cost. And we have taken a series of measures to prevent these risks.

Regarding identity information risks, our measures are as follows:

- We make a risk statement for participants and sign an identity information confidentiality agreement.

98   • We only use game statistics without identity information in our research .

99   • Special equipment and game accounts are provided to the participants to prevent leakage of
100    equipment and account information.

101   • The identity information of all participants is not disclosed to the public.

102 To compensate participants for their time cost, we offered each participant \$5 per match. Each match
103 is about 10-20 minutes, and participants can get about an average of \$20 an hour.

104 Finally, we have performed a process similar to IRB before the test is conducted. Our institution and
105 all participants have approved our research.

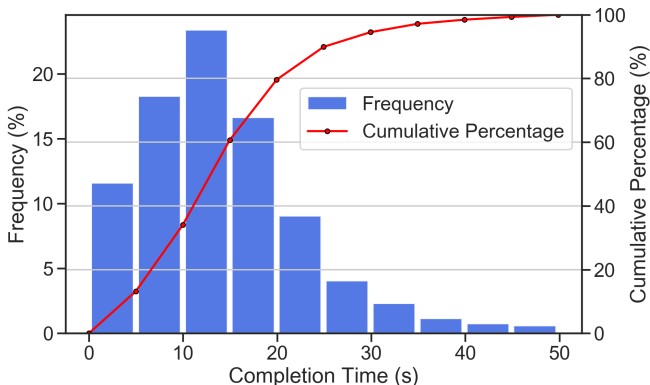

Figure 6: Time statistics of humans completing meta-commands in real games.

### A.10 Additional Experimental Results

#### A.10.1 CEN

**Training Data**. We extract meta-commands from expert game replay authorized by the game provider,
which consist of high-level (top 1% player) license game data without identity information. The
input features of CEN are shown in Table 4. The game replay consists of multiple frames, and the
information of each frame is shown in Figure 1. For setting $T^{mc}$, we counted the player's completion
time for meta-commands from expert game replay, and the results are shown in Fig. 6. We can
see that 80% meta-commands can be completed within the time of 20 seconds in *Honor of Kings*.
Thus, $T^{mc}$ is set to 300 time steps (20 seconds). Given a state $s_t$ in the trajectory, we first extract
the observation $o_t$ for each hero. Then, we use a hand-crafted command extraction function $f^{ce}$ to
extract the meta-command $m_t = f^{ce}(s_{t+T^{mc}})$ corresponding to the current state $s_t$ in the future.
By setting up labels in this way, we expect the CEN $\pi_\phi(m|o)$ to learn the mapping from the current
observation $o_t$ to its meta-command $m_t$. The detailed training data extraction process is as follows:

• First, we extract the trajectory $\tau = (s_0, \ldots, s_t, \ldots, s_{t+T^{mc}}, \ldots, s_N)$ from the game replay, where
 $N$ is the total number of frames.

• Second, we randomly sample some frames $\{t|t \in \{0, 1, \ldots, N\}\}$ from the trajectory $\tau$.

• Third, for each frame $t$, we extract feature $o_t$ from state $s_t$.

• Fourth, we extract the label $m_t$ from the state $s_{t+T^{mc}}$ in frame $t + T^{mc}$, i.e. describe the state
 using the meta-command space $M$.

• Finally, $< o_t, m_t >$ is formed into a training pair as a sample in the training data.

**Optimization Objective**. After obtaining the dataset $\{< o, m >\}$, we train the CEN $\pi_\phi(m|o)$
via supervised learning (SL). Due to the imbalance of samples at different locations of the meta-
commands, we use the focal loss [3] to alleviate this problem. Thus, the optimization objective
is:

$$L^{SL}(\phi) = \mathbb{E}_{O,M}\left[-\alpha m(1 - \pi_\phi(o))^\gamma \log(\pi_\phi(o)) - (1 - \alpha)(1 - m)\pi_\phi(o)^\gamma \log(1 - \pi_\phi(o))\right],$$

where $\alpha = 0.75$ is the balanced weighting factor for positive class ($m = 1$) and $\gamma = 2$ is the tunable focusing parameter. Adam[2] is adopted as the optimizer with an initial learning rate of 0.0001.

**Experimental Results**. Figure 7 shows the meta-command distributions of the initial CEN, the converged CEN, and strong humans. We see that the meta-commands predicted by the CEN gradually converge from chaos to the meta-commands with important positions. And the distribution of the converged CEN in Figure 7(b) is close to the distribution of strong humans in Figure 7(c) and the corresponding KL divergence is 0.44, suggesting that the CEN can simulate the generation of human meta-commands in real games.

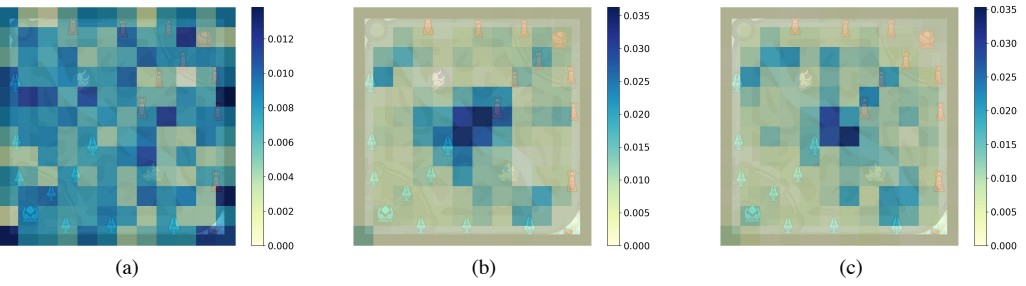

Figure 7: **The meta-command distributions of CEN and strong humans.** (a) The meta-command distribution of the initial CEN. (b) The meta-command distribution of the converged CEN. (c) The meta-command distribution of strong humans

### A.10.2 MCCAN

**Optimization Objective.** The MCCAN is trained via goal-conditioned RL with the goal of achieving a near-human completion rate for the meta-commands generated by the pre-trained CEN while ensuring that the win rate is not reduced. We adopt an intrinsic reward to guide the process of executing the meta-command $m_t$:

$$r_t^{int}(s_t, m_t, s_{t+1}) = \left| f^{ce}(s_t) - m_t \right| - \left| f^{ce}(s_{t+1}) - m_t \right|,$$

where $f^{ce}$ is a hand-crafted command extraction function. We train the MCCAN with the objective of maximizing the expectation over extrinsic and intrinsic discounted total rewards:

$$G_t = \mathbb{E}_{s \sim d_{\pi_\theta}, a \sim \pi_\theta} \left[ \sum_{i=0}^{\infty} \gamma^i r_{t+i} + \alpha \sum_{j=0}^{T^{mc}} \gamma^j r_{t+j}^{int} \right],$$

where $\alpha$ is a trade-off parameter and $d_\pi(s) = \lim_{t \to \infty} P\left(s_t = s \mid s_0, \pi\right)$ is the probability when following $\pi$ for $t$ steps from $s_0$.

**Training Process.** The MCCAN is trained by finetuning a pre-trained micro-action network [5] conditioned on the meta-command sampled from the pre-trained CEN. We modified the Dual-clip PPO algorithm [5] to introduce the meta-command $m$ into the policy $\pi_\theta(a_t | o_t, m_t)$ and the advantage estimation $A_t = A(a_t, o_t, m_t)$. The Dual-clip PPO algorithm introduces another clipping parameter $c$ to construct a lower bound for $r_t(\theta) = \frac{\pi_\theta(a_t | o_t, m_t)}{\pi_{\theta_{old}}(a_t | o_t, m_t)}$ when $A_t < 0$ and $r_t(\theta) \gg 0$. Thus, the policy loss is:

$$L^\pi(\theta) = \mathbb{E}_{s,m,a}[\max(cA_t, \min(\text{clip}(r_t(\theta), 1 - \tau, 1 + \tau)A_t, r_t(\theta)A_t)],$$

where $\tau$ is the original clip parameter in PPO. And the multi-head value loss is:

$$L^V(\theta) = \mathbb{E}_{s,m}[\sum_{head_k} (G_t^k - V_\theta^k(o_t, m_t))], V_{total} = \sum_{head_k} w_k V_\theta^k(o_t, m_t),$$

where $w_k$ is the weight of the $k$-th head and $V_t^k(o_t, m_t)$ is the $k$-th value.

**Experimental Results**. We conducted experiments to explore the influence of the extrinsic and intrinsic reward trade-off parameter $\alpha$ on the performance of MCCAN, and the win rate and completion rate results are shown in Figure 8. We see that as $\alpha$ increase, the completion rate of MCCAN

gradually increases, and the winning rate of MCCAN first increases and then decreases rapidly. When $\alpha = 16$, the completion rate of the trained agent for meta-commands is 82%, which is close to the completion rate of humans (80%). And the win rate of the trained agent against the SOTA agent [1, 5] is close to 50%. Thus, we finally set $\alpha = 16$ in subsequent experiments.

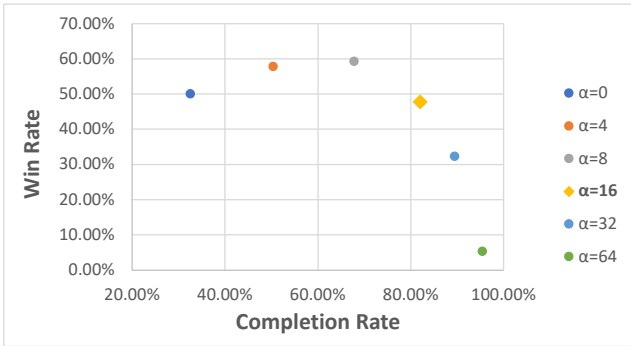

Figure 8: The win rate and completion rate of MCCAN with different $\alpha$. The opponent is the pre-trained action network, i.e., MCCAN with $\alpha$=0.

Table 7: The WRs of different strong human-AI teams against the MC-Base agents in *m AI + n Human* mode.

| Team Mode | Type of Agent | | |
|---|---|---|---|
| | MC-Base | MC-Rand | MCC |
| 2 AI + 3 Human | 8% | 3% | **18%** |
| 3 AI + 2 Human | 26% | 18% | **39%** |
| 4 AI + 1 Human | 42% | 28% | **54%** |

### A.10.3 Human-AI Game Test

**m AI + n Human Mode Result.** In addition to validating the generalization of the MCC agents to different levels of human teammates, we also evaluated the performance of the MCC agents to different numbers of human teammates. We had different numbers of strong humans team up with different types of agent teammates in *m AI + n Human* mode, where $m + n = 5$. We tested three team modes, including *2 AI + 3 Human* mode, *3 AI + 2 Human* mode, and *4 AI + 1 Human* mode. The corresponding WR results are shown in Table 7. We can see that as the number of humans increases, the WR of the MC-Base-Human team drops dramatically as expected. Note that the WR of the SOTA [1, 5] agent-only team against the human-only team is close to 100% and the WR of the MC-Base agent against the SOTA agent is close to 50%(see in Fig. 8) Fortunately, when humans team up with MCC agents, they can achieve effective communication and collaboration on macro-strategies, resulting in significant increased WRs. We can also see that when humans team up with MC-Rand agents, the WR is the lowest, suggesting that randomly communicating and collaborating can greatly hurt performance.

Table 8: The subjective preference results of all participants in the Human-AI Game Test.

| Participant Preference Metrics (from poor to perfect, 1~5) | Teammate | Type of Agent | | |
|---|---|---|---|---|
| | | MC-Base | MC-Rand | MCC |
| Reasonableness of H2A | Average Human | 2.3 ± 0.38 | 2.7 ± 0.24 | **4.0 ± 0.6** |
| | Strong Human | 2.2 ± 0.21 | 2.5 ± 0.41 | **4.1 ± 0.55** |
| Reasonableness of A2H | Average Human | - | 1.9 ± 0.35 | **4.3 ± 0.31** |
| | Strong Human | - | 1.7 ± 0.24 | **4.4 ± 0.35** |
| Overall Preference | Average Human | 2.7 ± 0.41 | 1.3 ± 0.27 | **4.3 ± 0.4** |
| | Strong Human | 2.5 ± 0.21 | 1.2 ± 0.17 | **4.5 ± 0.41** |

**Subjective Preference Results.** During the Human-AI Game Test, after completing each game test, the testers gave scores on several subjective preference metrics to evaluate their agent teammates, including the Reasonableness of H2A (how well agents respond to the meta-commands sent from testers), the Reasonableness of A2H (how reasonable the meta-commands sent from agents), and the Overall Preference for agent teammates. We separate the scores of strong humans and average humans and present the results in Table 8. We can see that for the Reasonableness of H2A metric, both strong and average humans gave the highest scores to MCC agents, which are significantly higher than that of other agents, indicating that humans relatively agree with the value estimation of MCC agents on meta-commands sent from humans. This is also verified in Fig. 7 of the main text. We can also see that for the Reasonableness of A2H metric, humans rated MCC agents much better than MC-Rand agents, indicating that humans believe that the meta-commands sent from MCC agents are more aligned with their own value system, so humans are more willing to trust and collaborate with MCC agents. For the Overall Preference metric, humans are satisfied with teaming up with MCC agents, scoring the highest scores compared to other agents. The results of these subjective preference metrics are also consistent with the results of objective metrics (Tables 1,2 of main text and Table 7 of Appendix).

## A.11 Limitations and Future work

### A.11.1 Limitations

There are three main limitations to our research work. 1) Due to the complexity of the MOBA game and the complexity of the MCC framework, the MCC framework adopts a sequential training manner instead of an end-to-end training manner. Thus, the training process of the MCC framework is tedious. 2) The training of the MCC agent consumes a lot of computing resources like the training of the SOTA MOBA AI agent. Thus, the computational cost of extending the MCC framework to other complex MOBA games is huge. 3) The meta-command we proposed is generic only to MOBA games and cannot be directly extended to other types of games, such as First-Person Shooting (FPS) and Massively Multiplayer Online (MMO).

### A.11.2 Future work

**From the application side**, we will precipitate this research work and apply it to the friendly bots in teaching mode of *Honor of Kings*, aiming to provide gameplay teaching to novice players.

**From the research side**, first of all, we will optimize the training process of the MCC framework, including the training process of the SOTA AI systems, reduce the computing resources required for training the MOBA agent, aiming to lower the threshold for researchers to study and reproduce work on MOBA games. Second, we will design a more general meta-command representation, such as natural language, and extend the MCC framework to other types of games. All in all, it is our sincere hope that human-AI collaboration in complex environments will attract more and more researchers' attention, and we also hope that this work can provide researchers with some new ideas.