# OpenReview forum: "Towards Effective and Interpretable Human-AI Collaboration in MOBA Games"
_NeurIPS.cc/2022/Conference — NeurIPS 2022 Submitted_

### Official Review · Reviewer_xzzg · 2022-07-10

**Rating:** 3
**Confidence:** 3
**Ethics Flag:** Yes
**Soundness:** 2 fair
**Presentation:** 3 good
**Contribution:** 2 fair

**Summary:**

The paper describes a system which is able to collaborate with human players in MOBA for the communication, selection, and execution of mid-level tasks (20s). They propose a simple communication method which is used by agents and human players, which is translated to an internal representation, evaluated in comparison with other goals, and executed using 3 different systems. Few details of the training sets are provided. The system is evaluated in comparison with other weaker strategies and is only about 10% better than an agent which does not communicate (MC-Base). In tests with human players, insufficiently described, it shows a good performance when paired with strong humans.

**Questions:**

1. Where does the training data come from? How much data? Were the users informed of the use of the data for training an AI system? Did they consent?
2. Did you access the ability level of the gamers provided by the game company?
3. Why did you not compare human-human results with human-ai results?
4. Why did MC-Base performed so well, and why this was not properly discussed in the paper?

**Ethics Review Area:**

["Inadequate Data and Algorithm Evaluation", "Inappropriate Potential Applications & Impact  (e.g., human rights concerns)"]

**Limitations:**

As I mentioned in the S&W section, the authors tend to overstate their claims, especially considering that: (1) they are only considering mid-level instructions; (2) they did not compare to a human baseline; (3) they do not know if real human players would use their MCC framework in real-stake games; and (4) there is no evidence that the populations they tested are representative.

**Strengths And Weaknesses:**

Strengths:
The paper is ambitions and, to a certain extent, complete, in the sense of aiming to solve an entire problem which is complex and, to some extent, hard to define, develop, and evaluate adequately.

The results of the paper (the ones which are acceptable, see Weaknesses) are promising, although since there is no baseline with other human players, it is hard to evaluate the actual performance level of the system which was developed.

The most important contribution of this paper, in my view, is providing some evidence that massive use of NNs and GPUs, in supervised learning, may be able to create collaborative AI players which understand mid-level instructions in complex game scenarios.


Weaknesses:

The paper is hard to read, mostly because there is an excessive effort to include everything in the main text. It comprises significant amount of work and hardware training, but it takes a very careful examination and reading to understand it. The paper is not clear about what is in the appendix (for instance, the details of the human experiment), so the reader has to mine things as it reads along. It was a frustrating experience to read this paper.

A key problem of the paper is that it exaggerates its claims. For instance, the MCC framework is very limited as a communication strategy, since it takes care only of mid-level strategy (20s). Collaboration ranges from quick commands to complex strategies which cover the whole game. That issue is never discussed in the game, and the proposal of the MCC framework seems to have never considered how human players actually communicate and interact in a real game. MCC seems to have been designed for easy of translation.

The authors also use the term interpretable in a way that is confusing with the current use in the field. While the Machine Learning field uses the term often to convey ML models whose behavior can be understood by human beings, in this case it means commands, issues by either machines or humans, in a subset of human language. At best, it is a confusing use, at worst, it is misleading.


There are some key methodological flaws in the paper. The main one, a very important one, is related to where the data used to train the system comes from, how much data is used, and etc.

It is not clear why the authors did not compare with the WR of only human teams, which would provide some sort of baseline to the results. Also, the analysis of the Agent-Only tests is too limited and, in many ways, misleading. Figure 6 shows that the MC-Base system is almost as good as MCC, which raises questions of how effective and important communication is in the evaluation task used in the paper. In Figure 5, we see that clearly in the under-60% win rate against MC-Base. This is an important result which is ignored in the paper, misrepresenting, at some extent, how good the results are. Papers should be as clear about what did not work as about what worked, which is clearly not this case.

Moreover, there is so little information about the players used in the test which is hard to trust the results. The authors blindly accepted the game makers' classification and abilities of the players, which could be simply wrong.

This paper also raises some ethical concerns. The authors never described where the training data comes from. Is it from real users' data? In this case, where they informed and consented to have their data used this way? Also, collaborative and communicative AI players raise concerns of cheating and of commercial exploitation of money-based platforms. They authors ignore those issues.

Finally, I found it problematic when the authors claim that they are the first to investigate the HAC problem in MOBA games, especially considering that this kind of research dates back to the 90s. In particular, the authors should look into David Chapman's pioneering work in the field:
* Vision, Instruction, and Action by David Chapman , The MIT Press 1991
* Agre, Philip E., and David Chapman. "What are plans for?." Robotics and autonomous systems 6.1-2 (1990): 17-34.

---

> ### Author Response · Authors · 2022-08-01
> **Response to Reviewer xzzg (1/2)**
>
> Thank you for the thorough and constructive comments. We provide clarification to your questions and concerns below. If you have any further questions or comments, we will be happy to have further discussions.
>
> **Q1: The paper is not clear about what is in the appendix.**
>
> A1: Because the content in Appendix was not complete when we submitted the paper, the subsection corresponding to Appendix was not clearly indicated in the paper. We have revised this problem and submitted the rebuttal revision version.
>
>
> **Q2: The MCC framework takes care only of mid-level strategy (20s). Collaboration ranges from quick commands to complex strategies which cover the whole game is never discussed.**
>
> A2: As stated in Line 150-151 (Main Text), $T^{mc}$ is the Time Limit for executing the meta-command. The MCC framework actually supports quick meta-commands. Besides, we counted the human completion time for meta-commands from expert data authorized by the game provider, and the results are shown in [Figure 6](https://sites.google.com/view/mcc-demo/%E9%A6%96%E9%A1%B5#h.avux32b91yfw). We can see that 80% of meta-commands can be completed within the time of 20 seconds in Honor of Kings. The frequency of the "long-term" meta-command is very small, and these "long-term" meta-commands are also difficult to model due to the credit assignment problem.
>
> Thus, $T^{mc}$ is set as 300 time steps (20 seconds) during the MCC training process. In the Human-AI Game Test, the testers can send a meta-command at any frequency, and the MCC agents will judge the value of the meta-command sent from testers through CS immediately to decide whether to execute it.
>
> **Q3: Use the term interpretable in a way that is confusing with the current use in the field.**
>
> A3: The interpretability of the MCC framework is reflected in two aspects: **(1) The meta-command itself is interpretable.** The signaling system is the most important and direct way to communicate macro-strategies between teammates in MOBA games. Our proposed meta-commands are based on the signaling system, and the MCC framework supports the conversion between meta-commands and signals. **(2) The collaboration between human players and agents is interpretable.** Human players can decide whether to execute the meta-commands sent from agents according to their own value system. MCC agents can also judge the value of meta-commands sent from humans through CS to decide whether to execute them. Figure 7 (Main text) shows that the value estimation results of CS are consistent with the ranking results of strong humans, confirming the interpretability of the collaboration behavior between MCC agents and humans.
>
> **Q4: Where does the training data come from? How much data? Were the users informed of the use of the data for training an AI system? Did they consent?**
>
> A4: For the training process of the MCC framework, only the CEN network needs to be trained with human data. The training data we used is high-level (top 1% player) license game data provided by the game developer with all player personal information stripped. We have also signed an agreement with the game developer to ensure that these data are only used for scientific research and not for any profit-making activities. In addition, we also attach great importance to ethical issues during human-AI game testing. For a detailed description of this, please see Appendix A.9.
>
> **Q5: Did you access the ability level of the gamers provided by the game company?**
>
> A5: For MOBA games, the rank system is the main way to differentiate players of different skill levels. For the training of the CEN network, we use the game data of the top 1% of players provided by the game developers. In addition, we also cooperate with game developers to find testers who meet the rank-level and experience requirements to participate in the Human-AI Game Test. The game developer helped us to find 15 top1% and 15 top 30% testers who volunteered to participate. All testers' personal information is anonymized to us, and all testers are informed of ethical concerns before the test begins. For a detailed description of this, please see Appendix A.9.

---

> > ### Author Response · Authors · 2022-08-01
> > **Response to Reviewer xzzg (2/2)**
> >
> > **Q6: Why did you not compare human-human results with human-ai results?**
> >
> > A6: **First, we would like to clarify that MC-Base can be considered as SOTA.**  In Line 244-245 (Main text), we mentioned "The MCCAN is trained by finetuning a pre-trained micro-action network [1]". In fact, [1] is the WuKong agent, the SOTA agent in HoK. In Line 260-261 (Main text), we stated "the MC-Base agent (agent only executes its own meta-command without communication)". Thus, MC-Base = CEN + MCCAN ($\alpha=16$). Figure 8 (Appendix) shows that the WR of the MC-Base agent against the SOTA agent [1] is close to 50\%. Thus, **MC-Base can be considered as SOTA.**
> >
> > **Second, we would like to explain why there is no comparison with the human-only team on the WR metric.** For the WR metric, we think it might not make sense to compare the WR of the human-only team, since the MC-Base (SOTA) agent can easily beat the top human players [2,3]. In addition, because the AI agent is not subject to any external interference, the performance of the AI agent is more stable than that of humans. Therefore, in the Human-AI Game Test, we use the MC-Base agent-only team with relatively more stable performance as the opponent to test the metrics of the human-AI team more accurately and fairly. For the RR metric, Table 2 (Main text) presents the results of human-AI and human-human teams. We can see that the RR of the MCC agents (73.05%) is very close to the RR of strong humans themselves (74.91%).
> >
> > **Q7: Interpretation of MC-Base and MCC results in Figures 5 and 6**
> >
> > A7: First, we would like to explain the difference between the MCC results in Figures 5 and 6 (Main text). **The difference between the MCC results in Figures 5 and 6 is mainly due to the different experimental environments.** We mentioned at the beginning of Section 5 of the paper that the results in Figure 5 were trained and tested on the full hero pool (more than 100 heroes), see Table 1 (Appendix), while the results of the ablation experiments shown in Figure 6 are trained and tested on a small hero pool (only 20 heroes), see Table 1 (Appendix). Reinforcement learning is a learning process that gradually overfits the environment, thus it is easier to overfit on small hero pools than on large hero pools (far more complex), which is why the win rate in Figure 6 is relatively high.
> >
> > Second, we would like to clarify the difference between the MCC results and the MC-Base results in Figures 5 and 6 (Main text). In fact, the MC-Base agent already has the level of top human players. Therefore, it is already a clear advantage to increase the win rate by about 10% on this basis. And this improvement is only brought about by effective communication and collaboration between agents. Note that, in Line 260-261 (Main text), we stated "the MC-Base agent (agent only executes its own meta-command without communication)"
> >
> > All in all, the results of Figures 5 and 6 are consistent with our conclusion: **the MCC framework can improve the win rate for the collaboration between agents.**
> >
> > **Q8: It is problematic when the authors claim that they are the first to investigate the HAC problem in MOBA games**
> >
> > A8: What we state in the paper is that "we are the first to investigate the Human-AI Collaboration problem in MOBA games". There has been a lot of research in the field of human-AI collaboration over the past few decades, but our emphasis is **in MOBA games**.
> >
> > Besides, as you mentioned, "especially considering that this kind of research dates back to the 90s." According to the introduction of MOBA in the wiki [4], the MOBA genre was established in 2000 and the first known research-based MOBA AI agents were published around 2015 for League of Legends. The two articles you listed are enlightening, but there are differences with MOBA game AIs.
> >
> > **References**
> >
> > [1] Ye, Deheng, et al. "Towards playing full moba games with deep reinforcement learning." NeurIPS 2020.
> >
> > [2] Tencent's AI beats human players to win Honour Of Kings mobile game, https://www.thestar.com.my/tech/tech-news/2019/08/07/tencent039s-ai-beats-human-players-to-win-honour-of-kings-mobile-game
> >
> > [3] Honor of Kings “Wukong AI” 3:1 defeated the human team, https://www.sportsbusinessjournal.com/Esports/Sections/Technology/2021/07/HoK-AI-Battle.aspx?hl=KPL&sc=0
> >
> > [4] Artificial Intelligence in MOBAs, https://en.wikipedia.org/wiki/Multiplayer_online_battle_arena

---

> > > ### Comment · Reviewer_xzzg · 2022-08-09
> > > **Thanks for the answers**
> > >
> > > Considering all answers and comments from the authors, I am now even more convinced that this paper needs serious rewriting to fit the NeurIPs format, or should find another venue. If important things are left out because of space limitations, the authors should focus the paper on what is important and make sure the argument is correctly made there. As a reviewer, if proper connections to the Appendix were not made because it was not ready at the submission, it should be penalized. Submissions are evaluated based on what they are and not on what they could be. It is currently a confusing paper with serious writing issues and I do not see a safe way to guarantee it will be better in the final version.
> > >
> > >
> > > Regarding Q6, I understand (now) the difficulties to compare with the human-human players, given that the machine is much better than humans. So if I understand, this context is more like human beings driving powerful cars (which are much faster than human beings) compared to machines driving powerful cars. However, if is that so, it is hard to call such scenarios as "collaboration", in the traditional sense in the HCI and CSCW communities. This is more like a case of verbal interaction between human beings and machines. In other words, the whole framing of the paper as a case of collaboration is probably inadequate, reinforcing the need, in my opinion, of a thorough rewriting of the paper.
> > >
> > > Regarding Q4, even if the data is only used for scientific purposes, it is still necessary to obtain explicit consent from the users to avoid ethical issues.
> > >
> > > Regarding Q8, please do not consider Wikipedia as authoritative evidence of previous work. Chapman's work was absolutely ahead of its time.
> > >
> > > There is some good work here, but this paper does not seem to be ready for publication.

---

### Official Review · Reviewer_WAvp · 2022-07-11

**Rating:** 5
**Confidence:** 2
**Soundness:** 2 fair
**Presentation:** 2 fair
**Contribution:** 3 good

**Summary:**

The paper introduces an approach for effective  Human-AI collaboration in the MOBA game  Honor of Kings. This is achieved by developing the interpretable Meta-Command Communication (MCC) based framework as a bridge for effective and interpretable  Human - AI communication within the game.

**Questions:**

* What is the benefit the presented research contributes to the overall AI community given the lack of reproducibility?
* Do you plan to apply the method in parallel to some open environment and open-source the results? Like Dota2, for example, Dota2 is presented on Fig.1 , but the method has not been evaluated in the setting of Dota2, which is a bit confusing.
* Can the hand-crafted command extraction function be replaced with a learned function? to transfer the approach to the setting of different games.
* As I understand the evaluation of AI agents collaborating with Humans has been done based on the win rate. Are there any results evaluating the human player impression from pairing up with AI team-mates, i.e., is it easy to recognize a human team-mate from the AI team-mate?

**Limitations:**

I do not think that the limitations associated with the reproducibility of the research have been properly addressed. This includes access to the game training environment, trained checkpoints, etc..

**Strengths And Weaknesses:**

The work follows the line of new AI approaches development within the setting of MOBA computer games. The particular MOBA - the Honor of Kings is used as the environment in this work, HoK along with DOTA II appeared in a few preceding AI studies.

The paper presents the first Human-AI interaction study within a significantly more complex MOBA game environment than the preceding HAI state-of-the-art including the Hanabi, Overcooked, and Capture the Flag environments. To achieve the studied goal, the paper introduces an interpretable protocol for H2AI and AI2H communication - the meta-command communication engine. The experimental evaluation is extensive and well documented. The paper is generally well written, with many figures describing their approach, results from the experiment, and an ablation study.

On one hand it is hard to criticise the empirical novelty of the presented work, on the other hand, I object the clear lack of reproducibility of the work : the training/evaluation environment is not available, the datasets of the games used for meta-command training are not shared, agent checkpoints are not available as well,  and the overall training procedure required a formidable amount of compute.  Moreover, the method has been tested in the restricted setting of a single environment (Honor of Kings) with a pool of handcrafted meta-commands . I can see clear advantages of the work for the HoK player community, especially if the bot is deployed online, and available for online play/ practice. However, I have a hard time noticing clear contributions of the work to the overall (multi-agent) AI community.

I would like the authors to think about and present their point of view on what overall benefit their research provides for the overall AI community, which goes beyond the community of HoK players in light of the apparent lack of reproducibility of the method.

Ideally, as the environment and source code are proprietary, I would like to see the method being applied in parallel to an open environment, e.g. Dota II open API, so other researchers can test it and reproduce it in their own research.

To conclude, I'm on the fence about accepting the paper, with the main disadvantage being the lack of reproducibility of the whole approach. If the AC confirms that it is fine to present the results obtained using proprietary code, without the possibility of reproducing I will be fine accepting the paper.

---

> ### Author Response · Authors · 2022-08-01
> **Response to Reviewer WAvp**
>
> Thank you for this thoughtful review! We largely agree with the point that the reproducibility of research work is critical to the development of the AI community. Below we present our views and efforts for the overall AI community, hoping to address your concerns.
>
> **Q1: What is the benefit the presented research contributes to the overall AI community given the lack of reproducibility?**
>
> A1: We agree that the reproducibility of research work is critical to the development of the AI community, so we are actually doing our best to promote the open-source of the MOBA game environment and related AI system codes. As one of the most popular MOBA games in the world, the open-source of the HoK game environment is in conflict with its commercial value. However, due to the characteristics of multi-agent cooperation and competition, partial observability, and complex state-action space, MOBA games represented by HoK are naturally suitable as a research and testing environment for advanced AI technologies. Therefore, we have been committed to communicating with game developers and introducing them to the potential scientific research value of the HoK environment, hoping that they can provide an open platform for researchers to test advanced AI technologies. Fortunately, we have cooperated with several parties including game developers to promote **the open-source of the 1v1 mode environment of HoK [1] and related AI training framework**, and we are now working on the open-source of the 3v3 mode environment. In addition, we also **jointly held the HoK AI competition with several parties including game developers in the 31st FISU World University Games [2]**, inviting more than 100 teams from all over the world to participate, providing game environments and free computing resources for each team to reproduce and improve advanced MOBA AI technologies.
>
> In the future, we will make further efforts to provide a entirely free MOBA open source environment and related AI systems to promote the development of the overall AI community.
>
> **Q2:  Do you plan to apply the method in parallel to some open environments and open-source the results? Like Dota2.**
>
> A2: In fact, we have tried to apply the MCC framework to Dota2, but we have not yet obtained an official authorization to obtain an open source dataset of high-level player game data to train the CEN network. Different MOBA games have similar key components, see Figure 1 in the main text. The signaling system (Appendix A.2) is one of the main ways that all MOBA game developers have designed for macro-strategies communication between teammates. The meta-commands we proposed are based on the signaling system in MOBA games, which can easily realize the conversion between meta-commands and signals, so we believe that the cost of applying the MCC framework to the existing Dota2 AI system is relatively small.
>
> **Q3: Can the hand-crafted command extraction function be replaced with a learned function? to transfer the approach to the setting of different games.**
>
> A3: The signaling system is the most important and direct way for teammates to communicate macro-strategies in MOBA games. The meta-command communication protocol we designed is based on the signaling system, which can be easily applied to other MOBA games. In the future, we will design a more general meta-command representation, such as natural language, and extend the MCC framework to other types of games, such as First-Person Shooters (FPS) and Massively Multiplayer Online (MMO).
>
> **Q4: Are there any results evaluating the human player impression from pairing up with AI team-mates?**
>
> A4: In the Human-AI Game Test, we only show the objective metrics: the WR and the RR. In fact, during the Human-AI Game Test, after completing each game test, the testers gave scores on several subjective preference metrics to evaluate their agent teammates, including the Reasonableness of H2A (how well agents respond to the meta-commands sent from testers), the Reasonableness of A2H (how reasonable the meta-commands sent from agents), and the Overall Preference for agent teammates. We present the results of and discussion on human subjective preference metrics [here](https://sites.google.com/view/mcc-demo/%E9%A6%96%E9%A1%B5#h.5drjm4dzsjyw) and included these results in Appendix A.10.3. [Table 8](https://sites.google.com/view/mcc-demo/%E9%A6%96%E9%A1%B5#h.5drjm4dzsjyw) shows that **humans are satisfied with teaming up with MCC agents and gave the MCC agent the highest score on all three metrics**, which is consistent with the objective metrics results (Tables 1 and 2 in the main text and Table 7 in the appendix).
>
> **References**
>
> [1] Honor of Kings Game Environment, https://github.com/tencent-ailab/hok_env
>
> [2] 31st FISU World University Games, https://aiarena.tencent.com/aiarena/en/match/fisu

---

### Official Review · Reviewer_8yYn · 2022-07-12

**Rating:** 7
**Confidence:** 3
**Soundness:** 3 good
**Presentation:** 3 good
**Contribution:** 3 good

**Summary:**

In this paper the authors introduce an approach to allow for human-AI collaboration (HAC) in the game Honor of Kings. This is made possible by devising a meta-command communication (MCC)-based framework with three components. First, messages (human-agent or agent-agent) are converted to this meta-command representation, then the optimal meta-command is chosen, and finally a sequence of actions are selected conditioned on the optimal meta-command. The paper introduces this framework, how it is trained, and then gives the results of agent-agent and agent-human experiments.

**Questions:**

1. What is the authors’ motivation for this work?
2. How would the authors relate Hanabi hint tokens to their MCC framework?
3. How did the authors extract the training data for CEN?
4. Why didn’t the author’s collect data on human experience? Or if they did, why not report them?

**Limitations:**

As discussed above, there’s a number of technical limitations not discussed by the authors. Ethically, everything seems to be fine.

**Strengths And Weaknesses:**

In terms of strengths, the paper introduces a novel and detailed approach to human-AI collaboration/human-AI teaming that should extend to at least other MOBA games but could potentially serve as the basis for a general communication framework in domains with similar constraints. The paper itself, outside of some small language issues, is overall very well-written, and the results are fairly convincing. I have minor concerns but overall this seems to be a promising approach.

In terms of weaknesses, as I mentioned above, I have a number of minor concerns. First, there’s a surprising lack of discussion motivating the work presented in this paper. Essentially all readers are given is “We expect this work can be a foundation for future HAC research in complex environments.”. Some indication of planned use cases/applications would be appreciated. Are the authors thinking of using these agents as friendly bots in Honor for Kings? Or is this purely theoretical and meant to develop HAC approaches?

Second, while I largely agree with the author’s summary of related work, I think they undersell the extent to which Hanabi requires a simplified version of their MCC framework [1]. Given how human and AI agents most communicate using the “hint tokens”, there’s actually a fair amount of overlap between the frameworks. Some acknowledgement of this in the paper would be beneficial.

Third, there’s no discussion in the paper of the limitations of the MCC framework. For example, the authors give the definition of M as “M represents the space of interpretable messages, that is, the Meta-Commands in the MCC framework”. However, this space is not fully described in the paper or supplementary materials. Assumably, M would need to be rewritten to apply this framework to a different MOBA game, which would require significant human effort.

Fourth, related to the MCC framework, there’s no discussion of how the training data for CEN is extracted, only that “The training dataset {< o, m >} is obtained by extracting the observation o and its corresponding meta-command m from expert data.”. Some description of the methodology for extracting this dataset would be valuable. If it was done by hand this would greatly decrease the generality of this approach.

Fifth, there was no collection of data related to the experience of the human players, or at least there’s no reporting of said data. This seems to be a major omission and an odd choice for a human subject study design. The human player’s experience is an important aspect of any human-AI teaming. If the teaming is quantitatively effective but the humans didn’t like it, then that would negatively impact others attempting to adapt the approach.

Sixth and finally, the results with a single human are impressive, but from the supplementary material it’s clear there’s a major falloff when more than one human is included. This is an important result and runs counter to the author’s claims that “The experimental results show that the MCC agents can collaborate reasonably with human teammates and even generalize to collaborate with different levels and numbers of human teammates.”. I’d walk back these claims and include some acknowledgement of this falloff in the paper.

There were also some occasional issues with the language, for example “thus far simpler”-> “thus it is far simpler” and “achieving the collaboration with humans”-> “achieving effective collaboration with humans”.

Overall, I’m positive on this paper, but the number of minor issues above does temper my enthusiasm.


1. Siu, Ho Chit, et al. "Evaluation of human-AI teams for learned and rule-based agents in Hanabi." Advances in Neural Information Processing Systems 34 (2021): 16183-16195.

---

> ### Author Response · Authors · 2022-08-01
> **Response to Reviewer 8yYn (1/2)**
>
> We thank the reviewer for the thoughtful and constructive comments. Below we address each of the raised questions and concerns.
>
> **Q1: Discussion of the motivation.**
>
> A1: **From the application side**, as you mentioned, we will precipitate this work and apply this research work to the friendly bots in the teaching mode of Honor of Kings (HoK), aiming to provide gameplay teaching to novice players.
>
> **From the research side**, first of all, it is our sincere hope that human-agent collaboration in complex environments can attract more researchers' attention. Thus, we will optimize the training process of the MCC framework, including the training process of the state-of-the-art (SOTA) AI systems, reduce the computing resources required for training the MOBA agent, aiming to lower the threshold for researchers to study and reproduce work on MOBA games. Second, we will design a more general meta-command representation, such as natural language, and extend the MCC framework to other types of games, such as First-Person Shooters (FPS) and Massively Multiplayer Online (MMO).
>
> **Q2: How would the authors relate Hanabi hint tokens to their MCC framework?**
>
> A2: Thank you for recommending the paper (Siu, Ho Chit, et al., 2021) that expands our thinking on applying the MCC framework to other game genres. We will introduce its discussion in the main text. Below, we briefly state the idea of applying the MCC framework to Hanabi.
>
> We believe that Hanabi hint tokens can be thought of as meta-commands in the MCC framework that contains the sender's strategy. We can use human data to train CEN to simulate the hint policies of humans. Then, we can train an CS to choose the player that needs the hints. Finally, the MCCAN is trained to make decisions based on received hints.
>
> **Q3: Discussion of limitations, eg., the meta-command and the MCC framework.**
>
> A3: Due to space reasons, we put limitations in Checklist 1. The definition of the meta-command in Section 4.2 (Main text) is mainly designed for MOBA games. The signaling system (Appendix A.2) is the most important and direct way for teammates to communicate macro-strategies in MOBA games. The meta-command communication protocol we designed is based on the signaling system, thus the cost of applying the meta-command and the MCC framework to other MOBA games is relatively small. Our future work is not only to apply the MCC framework to other MOBA games, but also to design meta-commands into more general representations, such as natural language, and then apply them to different categories of games.
>
> **Q4: How did the authors extract the training data for CEN?**
>
> A4: We extract meta-commands from game replay authorized by the game provider, which are consist of high-level (top 1%) license data without identity information. The input features of CEN are shown in Table 4 (Appendix). The game replay consists of multiple frames, and the information of each frame is shown in Figure1 (Appendix). The detailed training data extraction process is as follows:
> - First, we extract the trajectory $(s_0, s_1, ..., s_N)$ from the game replay, where $N$ is the total number of frames.
> - Second, we randomly sample some frames {$t | t \in ${ $0,1,\dots,N$}} from the trajectory $\tau$.
> - Third, for each frame $t$, we extract feature $o_t$ from state $s_t$.
> - Fourth, we extract the label $m_t$ from the state $s_{t+T^{mc} }$ in frame $t+T^{mc}$, i.e. describe the state using the meta-command space $M$.
> - Finally, $<o_t, m_t>$ is formed into a training pair as a sample in the training data.
>
> Since meta-commands are generic to MOBA games, the above rules can easily be extended to new MOBA games.
>
> **Q5: Why didn’t the author’s collect data on human experience? Or if they did, why not report them?**
>
> A5: In the Human-AI Game Test, we only show the objective metrics: the WR and the RR. In fact, during the Human-AI Game Test, after completing each game, the testers gave scores on several subjective preference metrics to evaluate their agent teammates, including the Reasonableness of H2A (How well agents respond to the meta-commands sent from testers), the Reasonableness of A2H (How reasonable the meta-commands sent from agents), and the Overall Preference for agent teammates. Because the results of objective metrics have clearly demonstrated the effectiveness of the MCC framework and the space limitation reasons, we did not demonstrate the results of subjective metrics. We present the results of and discussion on human subjective preference metrics [here](https://sites.google.com/view/mcc-demo/%E9%A6%96%E9%A1%B5#h.5drjm4dzsjyw) and included these results in Appendix A.10.3. [Table 8](https://sites.google.com/view/mcc-demo/%E9%A6%96%E9%A1%B5#h.5drjm4dzsjyw) shows that **humans are satisfied with teaming up with MCC agents and gave MCC agents the highest score on all three metrics**, which is consistent with the objective metrics results (Tables 1 and 2 in the main text and Table 7 in the appendix).

---

> > ### Author Response · Authors · 2022-08-01
> > **Response to Reviewer 8yYn (2/2)**
> >
> > **Q6: Explanation of experimental results with different numbers of human teammates.**
> >
> > A6: **First, we would like to clarify that MC-Base can be considered as SOTA**.  In the detailed description of the MCCAN training (Line 244-245, Main text), we mentioned "The MCCAN is trained by finetuning a pre-trained micro-action network [1]". In fact, [1] is the WuKong agent, the SOTA agent in HoK. In Line 260-261 (Main text), we stated "the MC-Base agent (agent only executes its own meta-command without communication)". Thus, MC-Base = CEN + MCCAN (\alpha=16). As can be seen in Figure 8 (Appendix), the WR of the MC-Base agent against the SOTA agent [1] is close to 50\%. Thus, **MC-Base can be considered as SOTA**. The detailed training processes of CEN and MCCAN can be found in Appendix A.10.1 and A.10.2.
> >
> > **Second, we would like to introduce the performance of the MC-Base agent-human team**. At present, the SOTA agent can easily beat the top human players [2,3]. In the Human-AI Game Test, the opponent is the MC-Base agent-only team. So as the number of human players increases, the WR of the MC-Base agent-human team decreases. Fortunately, as can be seen from Table 1 (Main text) and Table 7 (Appendix), when humans team up with MCC agents, they can achieve effective communication and collaboration on macro-strategies, resulting in significant improvements in WR metric.
> >
> > **Q7: Occasional issues with the language**
> >
> > A7: We have revised this problem and submitted the rebuttal revision version.
> >
> > **References**
> >
> > [1] Ye, Deheng, et al. "Towards playing full moba games with deep reinforcement learning." NeurIPS 2020.
> >
> > [2] Tencent's AI beats human players to win Honour Of Kings mobile game, https://www.thestar.com.my/tech/tech-news/2019/08/07/tencent039s-ai-beats-human-players-to-win-honour-of-kings-mobile-game
> >
> > [3] Honor of Kings “Wukong AI” 3:1 defeated the human team, https://www.sportsbusinessjournal.com/Esports/Sections/Technology/2021/07/HoK-AI-Battle.aspx?hl=KPL&sc=0

---

> > > ### Comment · Reviewer_8yYn · 2022-08-08
> > > **Re: Response to Reviewer 8yYn (2/2)**
> > >
> > > Thank you for the detailed response and extra results, in particular around the human subjective response. It's my opinion that this has improved the paper and I have increased my rating from a weak accept to an accept.
> > >
> > > -Reviewer 8yYn

---

### Official Review · Reviewer_PjFP · 2022-07-12

**Rating:** 3
**Confidence:** 4
**Ethics Flag:** Yes
**Soundness:** 2 fair
**Presentation:** 2 fair
**Contribution:** 2 fair

**Summary:**

This submission takes on the challenge of human-agent collaboration in Honor of Kings, a MOBA (multi-player online battle arena) strategy game. It proposes a particular approach to human-agent collaboration based on an explicit communication framework (“meta-commands”) and a hierarchical approach to collaboration. The submission evaluates Meta-Command Communication, its proposed approach, in several multi-agent and human-agent experiments.

**Questions:**

Did participants provide informed consent for participating in this research and providing their data for research purposes?

How do state-of-the art agents such as WuKong perform in human-agent teams in Honor of Kings?

Do participants prefer playing on a team with the MCC agent over state-of-the-art agents (e.g., WuKong)?

Can the authors explain why human-agent collaboration methods developed for Overcooked (Carroll et al., 2019; Strouse et al., 2021) would fail in this setting?

See also the revision recommendations in the "Strengths and Weaknesses" section.

**Ethics Review Area:**

["Responsible Research Practice (e.g., IRB, documentation, research ethics)"]

**Limitations:**

I didn’t see any coverage of technical or conceptual limitations (e.g., how training sequentially might affect performance or whether the proposed method would work in non-MOBA collaborative domains). I would recommend that the authors discuss those areas, and reflect a bit on any potential negative societal impact. If they believe there is no risk, they should mention that.

**Strengths And Weaknesses:**

**First, an ethics flag: Did participants provide informed consent for their involvement in this research and for the use of their data toward this project?** The checklist states that the authors engaged a "process similar to IRB" and that participants signed a "risk statement" and "identity information confidentiality agreement", but it's not clear to me that they actually provided informed consent. Were they fully informed about the purpose of the research, did they confirm their approval for involvement, were they told about how to withdraw their data, could they report any misconduct or unexpected harms, etc.?

I appreciate the opportunity to review this work; it’s apparent that the authors put in a lot of effort to develop this human-agent collaboration framework and test it with human players.

I see several prominent strengths for this paper. It takes an interesting approach to human-agent collaboration, focusing on the role of explicit (bounded) strategic communication to help synchronize human and agent policies. I appreciate the use of both multi-agent and human-agent experiments to evaluate the effectiveness of the approach. And it’s terrific that the authors provide details for their experiments and approach in the supplemental information.

On balance, however, I think this work is not quite ready for acceptance. The paper is primarily weakened by insufficiencies in the experimental evaluations and description of the method. The latter may be somewhat straightforward to fix, but I imagine fixing the former may require some additional work.

The most pressing issues I can see lie with the agent evaluation. It’s great that the experiments include several baselines against which the MCC agent is compared. However, the baselines only include three variants of the MCC agent. This is a pretty weak benchmarking approach, since it effectively tests whether ablations decrease performance. The gold standard approach would be to test against other standard agents from this domain, including some baselines that operate completely independently from the meta-command framework. As mentioned at the beginning of the paper, prior research has developed state-of-the-art agents for Honor of Kings (e.g., WuKong; Ye et al., 2020a, 2020b). How do such state-of-the-art agents---even if developed for competition against human teams---perform when placed on teams with humans?

Relatedly, for space reasons, the main text reports results for only 4 agents-1 human games. Unfortunately, these are the least informative games for evaluating the human compatibility of agents, since win rate and other team metrics can be driven entirely by agents collaborating with each other. (Again, running human-agent experiments with state-of-the-art, non-MCC agents developed for all-agent team play would help establish useful reference points for the MCC agent’s performance here.)

Finally, recent human-agent interaction work emphasizes the importance of evaluating interactive agents on both objective performance metrics and subjective metrics like satisfaction. See Du et al. (2020, NeurIPS), Strouse et al. (2021, NeurIPS), and McKee et al. (2022, AAMAS). As discussed in these papers, recent conversations around research ethics and responsible development of AI substantially increase the importance of considering human autonomy and preferences in designing interactive agents. Can we really claim that an agent is “better” (or human-compatible at all) if people dislike interacting with it?

Aside from the evaluation approach, a few scattered observations on other issues:
- With regard to the field of human-agent collaboration, the paper claims more novelty than is warranted. In particular, it ignores work demonstrating effective human-agent collaboration in Overcooked from prior years at this conference: Carroll et al., (2019, NeurIPS) and Strouse et al. (2021, NeurIPS).
- Related to the prior point on baselines, the paper claims on page 3 that state-of-the-art AI agents “can only defeat human players but cannot collaborate well due to the communication gap between agents and humans”. Can the authors supply any evidence for the claim that OpenAI-Five, WuKong, etc. would fail with human teammates? (In particular, because of the communication gap between humans and agents?)
- It is unclear to me how several parameters are chosen (e.g., $T_{mc}$).
- I don’t understand the justification for the reward formula on line 196. Why is there a trade-off parameter? Why not just sum reward without that multiplication factor?
- Overall, there are some recurring issues with the manuscript style, particularly around statements of fact/necessity and unevidenced claims.
   - On the former, the manuscript often uses language implying that a claim is a fact or a necessary statement, rather than a proposal (“We propose [...]”). I’d recommend the authors be a bit more deferential toward alternative approaches to their methods. For example, one section of the paper claims that “in MOBA games, a macro-strategy comprises three components: where to go, what to do, and how long” (p. 4). If this is a definition inherited from another paper (“X et al. (2020) introduce the concept of “macro-strategies”, which they describe as…”) or that this paper is introducing (“We propose that a _macro-strategy_ consists of [...]”), that should be stated more directly so that readers can engage with those arguments. Otherwise, I would argue that this is an insufficient definition: What about _who_ is involved in the strategy? What about conditional strategies, which are only triggered by certain conditions or have branching paths? Similar problems occur elsewhere, such as the section on page 3 that asserts “the HAC problem in MOBA games can generally be divided in the Human-to-AI (H2A) and the AI-to-Human (A2H) scenarios”. What about symmetric communication scenarios, or scenarios where there is no communication at all? Again, the manuscript is a bit too assertive in making general claims.
   - Relatedly, there are a number of unevidenced and unsubstantiated claims throughout the manuscript. I’d gently encourage the authors to consider the duty we have to our readers and any researchers who will build from our work: we should reference prior work or offer empirical evidence when we make claims. What evidence supports the following claims?
     - “However, these AI systems can only defeat human players but cannot collaborate well due to the communication gap between agents and humans” (p. 3)
     - “However, these tasks do not consider long-term macro-strategies planning and collaborating” (p. 3)
     - “The HAC problem in MOBA games can generally be divided into the Human-to-AI (H2A) and the AI-to-Human (A2H) scenarios” (p. 3)
     - “It normally takes a human [...] 300 time steps (20 seconds)” “to complete a macro-strategy in MOBA games” (p. 4)
     - “Besides, humans always want their meta-commands to be responded to as much as possible” (p. 8)
     - A final question: the introduction suggests that human-agent collaboration is harder or distinct from multi-agent collaboration. Can the authors explain why this is the case?

I’d like to thank the authors again for their contributions to these research problems. The authors claim that this research is a first for the field of human-agent collaboration: if that’s the case, I think addressing these issues would do the field a service, setting solid groundwork for others who will follow in this area.

---

> ### Author Response · Authors · 2022-08-01
> **Response to Reviewer PjFP (1/2)**
>
> Thank you for the thorough and constructive comments. We provide clarification below for your questions and concerns, which we hope will lead to a positive change in our work. If you have any further questions or comments, we will be happy to discuss them further.
>
> ### Part 1：About Evaluation Approach
>
> **Q1: How do state-of-the-art (SOTA) agents such as WuKong perform in human-agent teams in Honor of Kings (HoK)?**
>
> A1: We would like to clarify two points.
>
> **First, MC-Base can be considered as SOTA**.  In Line 244-245 (Main text), we mentioned "The MCCAN is trained by finetuning a pre-trained micro-action network [1]". In fact, [1] is the WuKong agent, the SOTA agent in HoK. In Line 260-261 (Main text), we stated "the MC-Base agent (agent only executes its own meta-command without communication)". Thus, MC-Base = CEN + MCCAN ($\alpha=16$). Figure 8 (Appendix) shows that the WR of the MC-Base agent against the SOTA agent [1] is close to 50\%. Thus, **MC-Base can be considered as SOTA**.
>
> **Second, the WR of the MC-Base agent-human team**. At present, the SOTA agent can easily beat the top human players [2,3]. In the Human-AI Game Test, the opponent is the MC-Base agent-only team. So as the number of human players increases, the WR of the MC-Base agent-human team decreases. Fortunately, Table 1 (Main text) and Table 7 (Appendix) show that when humans team up with MCC agents, they can achieve effective communication and collaboration on macro-strategies, resulting in significant improvements in WR metric.
>
> **Q2: Are win rate and other team metrics driven entirely by agents collaborating with each other?**
>
> A2: In the Human-AI Game Test, the **only difference** between MCC with MC-Base is **the addition of human-to-agent and agent-to-human communication** (Line 275-281, Main text). For fair comparisons, we **disabled agent-to-agent communication**. Thus, **the interaction behavior between agents is the same as other compared agents**.
>
> As can be seen from Tables 1 and 2 (Main text) and Table 7 (Appendix), the results of the Human-AI Game Test reflect that the MCC agents can effectively collaborate with human teammates, resulting in significant improvements in WR and RR metrics.
>
> **Q3: Do participants prefer playing on a team with the MCC agent over state-of-the-art agents?**
>
> A3: Thanks for your suggestions on the subjective metrics. In the Human-AI Game Test, we only show the objective metrics: the WR and the RR. In fact, during the Test, the testers gave scores on several subjective preference metrics to evaluate their agent teammates, including the Reasonableness of H2A (How well agents respond to the meta-commands sent from testers), the Reasonableness of A2H (How reasonable the meta-commands sent from agents), and the Overall Preference for agent teammates.
>
> We will follow the analytical methods of subjective metrics in the papers Du et al. (2020, NeurIPS), Strouse et al. (2021, NeurIPS), and McKee et al. (2022, AAMAS), and include the results and discussion in Appendix A.10.3. Now, we present the discussion on these metrics [here](https://sites.google.com/view/mcc-demo/index#h.v7plrgb0t6wn). Table 8 shows that **humans are satisfied with teaming up with MCC agents and gave the highest score on all metrics**, which is consistent with the objective metrics results.
>
> **Q4: Can the authors explain why human-agent collaboration (HAC) methods developed for Overcooked would fail in this setting?**
>
> A4: First, [Table 9](https://sites.google.com/view/mcc-demo/index#h.n6onmi3xq0z) shows that HoK is far more complicated than Overcooked. Besides, compared to Overcooked, MOBA game developers provide an explicit message exchange mechanism, i.e., the signaling system, for the collaboration on macro-strategies between teammates, which is important for the game victory [4]. The differences in game mechanics impose great limitations on the scalability of these HAC methods. Second, these HAC methods implicitly model collaboration in the network without explicit communication between teammates, resulting in poor interpretability. Third, in Table 10, the SOTA [1] HoK AI model consists of about 170,000,000 parameters, thus maintaining a population of agents constitutes a prohibitive computational burden.
>
> **Q5: Did participants provide informed consent for participating in this research and providing their data for research purposes?**
>
> A5: We have detailed ethics descriptions in Appendix A.9. As stated in Appendix A.9.2, participants were given instructions before testing. In point 5 (Line 86-88), participants' game statistics will be only used for academic research, and participants can choose whether to participate or not. We also stated in Appendix A.9.3 that the risks of experiments are only leakage of identity information and time cost. We provide means to prevent the leakage of identity information and inform participants in advance. In addition, we also offer compensation for time spent.

---

> > ### Author Response · Authors · 2022-08-01
> > **Response to Reviewer PjFP (2/2)**
> >
> > ### Part 2：About Other Issues
> >
> > **Q6: Evidence for the claim that OpenAI-Five, WuKong, etc. would fail with human teammates? (In particular, because of the communication gap between humans and agents?)**
> >
> > A6: First, as can be seen from Tables 1 and 2 (Main text) and Table 7 (Appendix), as the number of human players increases, the WR of the MC-Base (can be considered as SOTA) agent-human team decreases. While the WR of the MCC agent-human team is significantly higher than that of MC-Base, confirming the effectiveness of the meta-commands communication between humans and agents. Note that the only difference between MCC with MC-Base is the addition of human-to-agent and agent-to-human communication.
> > Second, as shown in [Table 8](https://sites.google.com/view/mcc-demo/%E9%A6%96%E9%A1%B5#h.5drjm4dzsjyw), participants gave the MC-Base agent low scores for the Reasonableness of H2A and the Overall Preference metrics, indicating that the MC-Base agent rarely collaborates with human teammates, resulting in a poor team experience. Note that, there is no communication exists in the MC-Base agent-human team.
> >
> > **Q7: How to choose $T^{mc}$ ?**
> >
> > A7: The choice of $T^{mc}$ is stated in Line 153-155 (Main Text). We restate here that we counted the human's completion time for meta-commands from expert data authorized by the game provider and the results are shown in [Figure 6](https://sites.google.com/view/mcc-demo/%E9%A6%96%E9%A1%B5#h.avux32b91yfw). We can see that 80% of meta-commands can be completed within the time of 20 seconds in Honor of Kings. Thus, $T^{mc}$ is set to 300 time steps (20 seconds) during the MCC training process.
> >
> > **Q8: The trade-off parameter in the reward formula of CS on Line 196.**
> >
> > A8: This is related to the prior knowledge of MOBA games. We have a detailed explanation in Line 193-203 (main text). We restate here that the execution of a meta-command (macro-strategy) involves reaching the location and doing the event, which the latter is more important to the value of the meta-command. Thus, we introduce the parameter $\beta >1$ to indicate that doing an event is more important than reaching the location.
> >
> > ### Part 3：About Manuscript Style
> >
> > **Q9: Issues on the statements of fact/necessity**
> >
> > A9: We have revised the paper and included the necessary citations.
> >
> > **Q10: Issues on the unevidenced claims**
> > >"However, these AI systems can only defeat human players but cannot collaborate well due to the communication gap between agents and humans”
> >
> > A: As shown in A6. We have included relevant evidence in the paper.
> >
> > >“However, these tasks do not consider long-term macro-strategies planning and collaborating”
> >
> > A: As shown in A4. We have included relevant evidence in the paper.
> >
> > >“It normally takes a human [...] 300 time steps (20 seconds) to complete a macro-strategy in MOBA games”
> >
> > A: As shown in A7. We have included relevant evidence in the paper.
> >
> > >“Besides, humans always want their meta-commands to be responded to as much as possible”
> >
> > A: We have replaced this description with "Besides, the higher the agent's response rate to meta-commands, the more collaborative behaviors of the agent, thus we expect the response rate of CS as high as possible".
> >
> > **Q11: The introduction suggests that human-agent collaboration (HAC) is hard or distinct from multi-agent collaboration (MAC).**
> >
> > A11: We did not include the comparison between HAC and MAC in the Introduction. In our humble opinion, HAC is more difficult than MAC, especially in complex MOBA games. Collaboration between multiple agents can be implicitly modeled in the network, overfitting to collaborate behaviors through extensive training, easily falling into local optimums. When agents collaborate with humans, more instability, leads to worse performance of the agent. Besides, due to the lack of explicit communication gap, humans cannot understand the behavior of agents, resulting in lacking effective communication and collaboration between agents and humans.
> >
> > **Q12: Limitations and societal impact**
> >
> > A12: Due to space limitation reasons, we did not discuss the limitations and societal impact in the main text, and wrote them in Checklist 1. For example, in Line 480-483, we describe the "not verified in non-MOBA games" limitation; in Line 484-489, we also discuss the societal impact. We have added more detailed discussions on limitations and future work in Appendix A.11.
> >
> > **References**
> >
> > [1] Ye, Deheng, et al. "Towards playing full moba games with deep reinforcement learning." NeurIPS 2020.
> >
> > [2] Tencent's AI beats human players to win Honour Of Kings mobile game, https://www.thestar.com.my/tech/tech-news/2019/08/07/tencent039s-ai-beats-human-players-to-win-honour-of-kings-mobile-game
> >
> > [3] Honor of Kings “Wukong AI” 3:1 defeated the human team, https://www.sportsbusinessjournal.com/Esports/Sections/Technology/2021/07/HoK-AI-Battle.aspx?hl=KPL&sc=0
> >
> > [4] do Nascimento Silva, Victor, and Luiz Chaimowicz. "Moba: a new arena for game ai." arXiv 2017.

---

> > > ### Comment · Reviewer_PjFP · 2022-08-09
> > > **Thank you for your comments**
> > >
> > > Thank you to the authors for their reply to my comments. First, on the ethics issue, thank you to the authors for pointing to the additional methods information in the appendix. However, I don't think that "a process similar to IRB" is a sufficient amount of detail to provide on the review process. IRBs (and analogous bodies in other countries) are designed to act as independent ethical review, to provide continued oversight on human-participants research, and to offer a mechanism for participants to complain if there are issues. How does the process mentioned meet these standards? (E.g., the authors state that the risks of experiments are only leakage of identity information and time cost—but this appears to be their subjective assessment. Did an independent ethics review confirm this judgment?) If the process meets these standards, then more information should be provided in the paper.
> > >
> > > On the non-ethics points, the authors’ responses provide more detail on several of the problems identified; in some of those cases, they additionally edited the manuscript to address the issues. However, assessing the revised manuscript holistically, I think the analysis and interpretation need to be substantially improved before this is ready for acceptance at NeurIPS. At present, the evidence offered is not sufficient to persuasively establish the strong breakthrough in human-agent compatibility that the authors claim.
> > >
> > > Notes to the authors:
> > >  - Thank you for clarifying the claim that one of the baselines is the state-of-the-art agent in HoK. This claim is not evident (or as prominent as it should be) in the current text: the authors should make their case much more clearly than it currently is.
> > >  - The response to Q2 misunderstands the concern. Let's say we have three football players: player A, player B, and player C. We clone players B and C so that there are 10 of each. We then form two teams, A + 10B and A + 10C, and have them play a number of games against various other teams. Player B happens to play cooperatively with A, passing the ball and otherwise coordinating their strategy. A + 10B wins 60% of their games. Player C, on the other hand, has a strategy that plays exclusively with itself. The ten player Cs never pass the ball to A, but still play extremely effectively. A + 10C wins 90% of their games, despite player A never contributing to the game.
> > > As you can see, just because a team of AI agents has a high win rate when playing 1 human / 4 agents does not mean it is collaborative. In fact, without improved analysis, the patterns described in your paper could equally result from the MCC agent engaging in this approach.
> > >  - In response to A3: thank you for the additional information. Especially given the issues identified above in disentangling increases in win rate from actual collaborativeness, I think this needs some more thought and expansion. Did the participants provide verbal feedback on the games? More methodological information is necessary here to properly interpret the results (e.g., what is the wording of the questions? what scale is used?).
> > >  - I appreciate the comparison of game parameters between Overcooked and Honor of Kings. Nonetheless, the claim "The differences in game mechanics impose great limitations on the scalability of these HAC methods" feels strong given that there is no theoretical or empirical analysis to support it.
> > >  - Thank you for bringing the evidence in Q6 together in one place. This is a really interesting set of claims, but it is not well convened and articulated in the current version of the manuscript. I’d strongly encourage spending more time making the analysis and interpretation into a cohesive argument for human-agent compatibility: it’s a difficult topic in team games, and likely needs much more space than a half page (to disprove alternative explanations like excluding the human player, etc.).
> > >  - On A7, thank you for clarifying in the main text where this statistic comes from. Previously, it was stated without attribution.
> > >  - On A8, thank you for the explanation. This clarification should be included in the text.
> > >  - I think you did a good job addressing my concerns on A9 and A10. I am not as convinced by A11 and A12, but those concerns are minor relative to those identified around the evaluation, interpretation, and especially research ethics.

---

### Review · Ethics_Reviewer_Ei1K · 2022-07-27

**Recommendation:**

I agree with the two Reviewers who raised ethical issues. There is a possibility that the data were provided freely and with full knowledge of how the data would be used. There is also a possibility that the data were used without the users' agreement or consent. We need to know more about source and consent.

To the Authors:
Please review the ethical concerns raised by Reviewers PjFP and xzzg. Please add details of the source of training data and the consent from all users.

To All:
If possible, we should re-review the submission after the Authors have made these revisions.

**Ethical Issues:**

Yes

**Ethics Review:**

Two reviewers questioned the source of the training data and the nature of the consent given by users. These are real issues, which need ot be addressed.

---

### Author Response · Authors · 2022-08-01
**Response to All Reviewers**

Dear Reviewers,

We sincerely appreciate the thorough and constructive comments from reviewers. We are pleased to find that they find our approach interesting/novel/promising (PjFP, 8yYn, WAvp), well-written (8yYn, WAvp), the experiments comprehensive (PjFP, WAvp), and the results convincing/promising (WAvp, xzzg).

Although we have responded to each of you individually, we would like to clarify some key points of confusion and summarize the updates we have made to the manuscript.

**Confusion Points**

>The relationship between HoK SOTA agent and MC-Base agent

In Line 244-245 (Main text), we mentioned "The MCCAN is trained by finetuning a pre-trained micro-action network [38]". In fact, [38] is the WuKong agent (Ye et al., 2020), the SOTA agent in Honor of Kings. In Line 260-261 (Main text), we stated "the MC-Base agent (agent only executes its own meta-command without communication)". Thus, MC-Base = CEN + MCCAN ($\alpha=16$). Figure 8 (Appendix) shows that the WR of the MC-Base agent against the SOTA agent [38] is close to 50%. Thus, MC-Base can be considered as SOTA.

>Results of human subjective preferences

In the Human-AI Game Test, we only show the objective metrics: the WR and the RR. In fact, during the Human-AI Game Test, after completing each game test, the testers gave scores on several subjective preference metrics to evaluate their agent teammates, including the Reasonableness of H2A (how well agents respond to the meta-commands sent from testers), the Reasonableness of A2H (how reasonable the meta-commands sent from agents), and the Overall Preference for agent teammates. Because the results of objective metrics: the WR and the RR have clearly demonstrated the effectiveness of the MCC framework and the space limitation reasons, we did not demonstrate the results of subjective metrics.
We present the results of and discussion on human subjective preference metrics [here](https://sites.google.com/view/mcc-demo/%E9%A6%96%E9%A1%B5#h.5drjm4dzsjyw) and included these results in Appendix A.10.3.

>Ethics problem

We have detailed ethics descriptions in Checklist 5 and Appendix A.9.

>Discussion of limitations

Due to space reasons, we put the main limitation in Checklist 1. We restate in detail the limitations at [here](https://sites.google.com/view/mcc-demo/%E9%A6%96%E9%A1%B5#h.bxvba773k7de) and included limitations in Appendix A.11.

**Summary of The Updates**

With respect to the questions and suggestions all reviewers provided, we have done our best to improve the manuscript. Now, we would like to summarise the changes we have made to the manuscript:

1. Revision of the manuscript's language and statement.   ---- Line 74, 81, 113, 143, 155, 317 in the main text

2. Additional description of the MC-base and SOTA relationship.  ---- Line 245, 264, 285 in the main text

3. Additional descriptions of the CEN training data and its extraction process.  ---- Appendix A.10.1

4. Additional description of $T^{mc}$ settings.  ---- Appendix A.10.1 & Figure 6

5. Additional interpretation of Human-AI Game Test experimental results.  ---- Appendix A.10.3

6. Additional results and discussions of human subjective preferences.  ---- Appendix A.10.3 & Table 8

7. Additional limitations and future work.  ---- Appendix A.11

8. Additional discussions of related work  [Du et al. 2020], [Siu, Ho Chit, 2021], [McKee et al. 2022].

We thank you once again for your reviews and sincerely hope we could address your questions!

Paper 5064 Authors

---

### Public Comment · ~Shubao_Zhang2 · 2023-01-10
**Previous work of human-ai collaboration in MOBA games**

In 2021, Zhang studied the human-ai collaboration problem in MOBA games [1].

[1] Towards Controllable Agent in MOBA Games with Generative Modeling. arXiv:2112.08093v1, 2021.

---

### Meta-Review · Area_Chair_1nvj · 2022-08-25

**Recommendation:** Reject
**Confidence:** Certain

**Metareview:**

The paper proposes a mechanism for human-AI collaborative play in the game Honor of Kings. The high-level approach is sensible: communicating happens within a small space of "meta-commands", which can be converted to coherent chunks of agent behaviour. The complexity of the system is high, the experiments are done at large scale, and the empirical findings are impressive (in single-human collaboration at least). However, the reviewers also point out a large array of concerns to be resolved; of particular concern, and insufficiently addressed, are the ethical issues (informed consent) raised by multiple reviewers.

Overall, the reviewers find that the paper is not ready for publication yet, and I concur. I hope the authors will integrate the rich feedback from this reviewing process in a future (extensively rewritten) iteration.

**Award:**

No

---

### Decision · Program_Chairs · 2022-09-14

Reject